# SHAPLEY-GUIDED UTILITY LEARNING FOR EFFECTIVE GRAPH INFERENCE DATA VALUATION

**Hongliang Chi[1], Qiong Wu[2], Zhengyi Zhou[2], Yao Ma[1]**

[1]Rensselaer Polytechnic Institute, Troy, NY, United States
[2]AT&T Chief Data Office, Bedminster, NJ, United States
{chih3@rpi.edu, qw6547@att.com, zz547k@att.com, may13@rpi.edu}

## ABSTRACT

Graph Neural Networks (GNNs) have demonstrated remarkable performance in various graph-based machine learning tasks, yet evaluating the importance of neighbors of testing nodes remains largely unexplored due to the challenge of assessing data importance without test labels. To address this gap, we propose Shapley-Guided Utility Learning (SGUL), a novel framework for graph inference data valuation. SGUL innovatively combines transferable data-specific and model-specific features to approximate test accuracy without relying on ground truth labels. By incorporating Shapley values as a preprocessing step and using feature Shapley values as input, our method enables direct optimization of Shapley value prediction while reducing computational demands. SGUL overcomes key limitations of existing methods, including poor generalization to unseen test-time structures and indirect optimization. Experiments on diverse graph datasets demonstrate that SGUL consistently outperforms existing baselines in both inductive and transductive settings. SGUL offers an effective, efficient, and interpretable approach for quantifying the value of test-time neighbors.

## 1 INTRODUCTION

Data valuation, the task of quantifying the value of individual data points for machine learning (ML) tasks, has gained significant attention in recent years. As ML models and datasets continue to grow in scale and complexity, understanding the contribution of each data point becomes crucial for fair compensation, dataset curation, and business strategy-making (Pei, 2020; Sim et al., 2022; DalleMule & Davenport, 2017). Concurrently, Graph Neural Networks (GNNs) (Kipf & Welling, 2016; Veličković et al., 2017; Wu et al., 2019; Klicpera et al., 2018) have demonstrated remarkable performance in various graph-based machine learning tasks, including social network analysis (Fan et al., 2019b; Goldenberg, 2021), recommendation systems (Fan et al., 2019a; Xu et al., 2020), and molecular property prediction (Li et al., 2022; Wu et al., 2023). Notably, the success of GNNs, like other ML methods, heavily relies on informative data.

Unlike Euclidean data, graph data is characterized by its non-independent and identically distributed (non-IID) nature, where nodes influence each other according to the graph structure. This property necessitates specialized data valuation methods for graph data. Addressing this necessity, recent work has extended data valuation methods to graph data, introducing the concept of graph data valuation to assess the contribution of graph structures (Chi et al., 2024a). For general data valuation, game theoretic approaches (Ghorbani & Zou, 2019; Kwon & Zou, 2021; Wang & Jia, 2023; Chi et al., 2024a) have been widely adopted. As with other game theoretic methods, the value of a graph structure is derived from its marginal contributions to different subsets of the dataset. These marginal contributions are quantified by a utility function, which measures how the model's performance changes with or without a particular graph structure. Specifically, the utility function maps from a set of players (training graph structures) to their joint contribution to the model's performance, typically measured using accuracy on labeled validation nodes. Data values are then computed based on how the utility function's results change across different subsets of graph structures.

---

Part of this work was done while the first author was an intern at AT&T CDO.
Code is released at https://github.com/frankhlchi/infer_data_valuation.

Importantly, graph structures can contribute not only during training but also during inference, thus possessing inherent value. Recent work (Yang et al., 2023) has further emphasized the importance of graph structures at test-time, demonstrating that utilizing only testing graph information can achieve performance comparable to full GNNs. In real-world applications, such as GNN-based recommender systems, real-time recommendations often rely on dynamic graphs derived from social networks or item interactions. In these scenarios, assessing the value of inference-time graph data is crucial for optimizing user experience. While evaluating inference-time graph data is key for GNN performance improvement and real-world applications, most existing methods focus on training data valuation, leaving the value of inference graph structures, especially test node neighbors, largely unexplored.

However, designing data valuation methods for graph inference structures faces fundamental challenges: **1. Lack of test labels:** Traditional game-theoretic approaches rely on labeled validation data as a proxy for model performance on testing data. In the case of graph inference data valuation, accuracy on test nodes directly represents model performance. However, the ground truth labels are unavailable for test nodes. **2. Limitations of existing utility learning methods:** While surrogate models could potentially approximate utility functions for test-time structures, current methods (Wang et al., 2021) are inadequate for graph inference data valuation due to two primary issues: *(a) Inapplicability to unseen test-time structures:* Existing models are typically trained on mappings from training data subsets to validation accuracy. This approach fails when applied to a different set of players, such as test node neighbors. *(b) Indirect and inefficient optimization:* Current methods often use accuracy as the utility learning objective. This approach does not directly optimize the fitted Shapley values against ground truth Shapley values, which is the ultimate goal of data valuation. Furthermore, this indirect approach necessitates processing and storing large amounts of accuracy-level data, leading to increased memory requirements. This inefficiency becomes particularly problematic when dealing with large-scale graphs.

To address these challenges and enable effective graph inference data valuation, we propose Shapley-Guided Utility Learning (SGUL), a novel interpretable and efficient framework for estimating graph inference data value without relying on test labels. Our key contributions are as follows:

- We are the *first* to formulate the graph inference data valuation problem. This addresses a significant gap in the field, where the importance of test-time graph structures has been recognized but not quantified through data valuation methods.
- We propose SGUL, a novel utility-learning framework for graph inference data valuation that addresses the challenge of valuation without test labels. Our approach introduces a transferable feature extraction method that transforms player-dependent inputs into general features.
- We develop a Shapley-guided optimization method that enables direct optimization of Shapley values, improving computational efficiency and model effectiveness.
- We conduct extensive experiments on various graph datasets, demonstrating that SGUL consistently outperforms baseline methods in graph inference data valuation tasks.

## 2 PRELIMINARY STUDY

### 2.1 GAME-THEORETIC DATA VALUATION

Data valuation, the task of quantifying the contribution of individual data points to machine learning (ML) tasks, has gained significant attention in recent years. Game-theoretic approaches, particularly those based on cooperative game theory, have emerged as a prominent framework for data valuation (Ghorbani & Zou, 2019; Kwon & Zou, 2021; Wang & Jia, 2023). In this context, i.i.d training data points are treated as players in a cooperative game, where they can form coalitions to contribute collectively to model performance.

The Shapley value, a solution concept from cooperative game theory, has been widely adopted for fair allocation of value among players. For a player $i$ in a set of players $\mathcal{D}$, its Shapley value is defined as: $\phi_i(\mathcal{D}, U) = \frac{1}{|\Pi(\mathcal{D})|} \sum_{\pi \in \Pi(\mathcal{D})} [U(\mathcal{D}_i^\pi \cup \{i\}) - U(\mathcal{D}_i^\pi)]$ where $\Pi(\mathcal{D})$ is the set of all permutations of $\mathcal{D}$, $\mathcal{D}_i^\pi$ is the set of players that appear before $i$ in permutation $\pi$, and $U$ is the utility function. The Shapley value $\phi_i(\mathcal{D}, U)$ represents the average marginal contribution of data point $i$ across all possible coalitions. In practice, to reduce computational complexity, we often estimate Shapley values by sampling a subset of permutations rather than considering all possible permutations (Ghorbani & Zou, 2019; Jia et al., 2019). Central to these approaches is the utility function, which quantifies the

contribution of a subset of players to the overall performance. Formally, we can define the utility function as: $U : 2^{\mathcal{D}} \to \mathbb{R}$ where $2^{\mathcal{D}}$ represents the power set of $\mathcal{D}$. For any subset $S \subseteq \mathcal{D}$, $U(S)$ measures the performance achieved by this subset.

The representative work of Data Shapley (Ghorbani & Zou, 2019) first introduced Shapley value for data valuation. In this framework, as well as in subsequent game-theoretic approaches (Kwon & Zou, 2021; Wang & Jia, 2023), training samples are treated as players, and the utility function is typically defined based on the model's performance on a validation set: $U(S) = \text{Acc}(f_S, \mathcal{D}_{\text{val}})$ where $S \subseteq \mathcal{D}_{\text{tr}}$ is a subset of the training dataset, $f_S$ denotes a model trained on subset $S$, and $\mathcal{D}_{\text{val}}$ is a validation set used to evaluate the model's performance. This formulation allows us to quantify the contribution of different subsets of training data to the model's predictive accuracy.

## 2.2 GRAPH DATA VALUATION

While Data Shapley and subsequent methods (Ghorbani & Zou, 2019; Kwon & Zou, 2021; Wang & Jia, 2023) have been effective for i.i.d. data, they face significant challenges when applied to graph-structured data. A key challenge is capturing the hierarchical and dependent relationships among graph elements. In GNNs, a node's contribution to model performance is intricately linked to its position within the computation tree and its relationships with other nodes. Traditional Shapley value calculations, which treat all players (nodes) independently, fail to account for these crucial dependencies, potentially leading to inaccurate valuations. To address these challenges, recent work has proposed a more granular approach to graph data valuation (Chi et al., 2024a). This approach introduces two key constraints to the Shapley value calculation: the Level Constraint and the Precedence Constraint. These constraints capture the hierarchical and dependent relationships among graph elements. For a detailed discussion of these constraints, please refer to Appendix D.2.

## 3 METHODOLOGY

### 3.1 GRAPH INFERENCE DATA VALUATION PROBLEM

GNNs have demonstrated remarkable performance across various real-world applications, where the $k$-hop neighbors of key nodes provide essential context for predictions. For example, in protein-protein interaction networks, traffic systems, and social networks, neighboring nodes play vital roles in predicting properties of nodes of interests (Jha et al., 2022; Wang et al., 2022; Liu et al., 2021). Therefore, quantifying the importance of test-time neighboring structures becomes a critical challenge for improving GNN performance, particularly for applications requiring real-time inference.

Given these considerations, we formally introduce the graph inference data valuation problem:

**Input Setup:** (1) Training graph $\mathcal{G}_{\text{Tr}} = (\mathcal{V}_{\text{Tr}}, \mathcal{E}_{\text{Tr}}, \mathbf{X}_{\text{Tr}})$ with labeled nodes $\mathcal{V}^l_{\text{Tr}} \subset \mathcal{V}_{\text{Tr}}$ and labels $\mathbf{Y}_{\text{Tr}}$, and a fixed trained GNN model $f(\cdot)$; (2) Validation graph $\mathcal{G}_{\text{Val}} = (\mathcal{V}_{\text{Val}}, \mathcal{E}_{\text{Val}}, \mathbf{X}_{\text{Val}})$ with labeled nodes $\mathcal{V}^l_{\text{Val}}$ and labels $\mathbf{Y}_{\text{Val}}$; (3) Test graph $\mathcal{G}_{\text{Te}} = (\mathcal{V}_{\text{Te}}, \mathcal{E}_{\text{Te}}, \mathbf{X}_{\text{Te}})$ with target nodes $\mathcal{V}_t$ but no labels.

With this setup, we define the graph inference data valuation problem as follows:

**Definition 1** (Graph Inference Data Valuation). *Given a set of target test nodes $\mathcal{V}_t \subset \mathcal{V}_{Te}$, their neighborhood $\mathcal{N}(\mathcal{V}_t)$, and a downstream task $T$, the goal of graph inference data valuation is to learn a value-assignment function $\phi : \mathcal{N}(\mathcal{V}_t) \to \mathbb{R}$ that assigns scores to the neighbors of $\mathcal{V}_t$ based on their contribution to the performance of $f(\cdot)$ on task $T$.*

*Graph inference data valuation problem* primarily focuses on $\mathcal{N}_k(\mathcal{V}_t)$, the $k$-hop neighborhood of target nodes $\mathcal{V}_t$, where $k$ typically equals the number of GNN layers $L$. This choice naturally aligns with GNNs' message passing mechanism, ensuring we capture all nodes that influence target nodes' predictions through the network's receptive field.

### 3.2 STRUCTURE-AWARE SHAPLEY VALUES FOR GRAPH INFERENCE DATA VALUATION

To address the graph inference data valuation problem, we adopt the Structure-Aware Shapley value formulation with connectivity constraints (Chi et al., 2024a). Unlike PC-Winter (Chi et al., 2024a) which defines players as individual nodes in computation trees, our test-time valuation focuses solely on test node neighbors. This fundamental difference in player definition naturally leads us to use only

the Precedence constraint, which ensures connectivity between added nodes during value calculation, as the Level constraint from training-time valuation becomes inapplicable to our setting.

Formally, given a set of target test nodes $\mathcal{V}_t \subset \mathcal{V}_{\text{Te}}$ and their collective neighborhood $\mathcal{N}(\mathcal{V}_t)$, we define the *Structure-Aware Shapley value* for each neighbor as follows:

$$\phi_i(\mathcal{N}(\mathcal{V}_t), U) = \frac{1}{|\Omega(\mathcal{N}(\mathcal{V}_t))|} \sum_{\pi \in \Omega(\mathcal{N}(\mathcal{V}_t))} [U(\mathcal{N}_i^\pi(\mathcal{V}_t) \cup \{i\}) - U(\mathcal{N}_i^\pi(\mathcal{V}_t))] \qquad (1)$$

where $U : 2^{\mathcal{N}(\mathcal{V}_t)} \to \mathbb{R}$ is a utility function mapping from the power set of $\mathcal{N}(\mathcal{V}_t)$ (all possible subsets of neighbors) to real numbers, $\Omega(\mathcal{N}(\mathcal{V}_t))$ is the set of all permissible permutations that satisfy the graph connectivity constraints, and $\mathcal{N}_i^\pi(\mathcal{V}_t)$ is the set of neighbors that appear before $i$ in permutation $\pi$.

As shown above, the utility function $U(\cdot)$ is the most critical input for data value formulation. In traditional data valuation scenarios, as discussed in Section 2.1, the utility function typically maps subsets of training data to their performance on a validation set. For instance, $U(S) = \text{Acc}(f_S, \mathcal{D}_{\text{val}})$, where $f_S$ is a model trained on subset $S$, and $\mathcal{D}_{\text{val}}$ is a validation set. In our graph inference data valuation context, the utility function should ideally map subsets of test neighbors to their collective contribution to the GNN's performance on target nodes. While this formulation provides a theoretically sound approach to graph inference data valuation, we face a significant challenge: *for test nodes, the absence of ground truth labels makes it impossible to directly measure accuracy and define an appropriate utility function.*

## 4 SHAPLEY-GUIDED UTILITY LEARNING (SGUL)

As shown in the prior section, the critical challenge in solving the graph inference data valuation problem lies in obtaining an appropriate utility function without testing labels. To address this issue, we propose an approach to learn the utility function that can effectively value test node neighbors.

### 4.1 UTILITY LEARNING IN DATA VALUATION

Despite the crucial role of utility functions in data valuation, as introduced in Section 2.1, accessing these functions can often be costly or even impossible. For instance, the utility function we previously discussed for training data valuation requires model retraining, which can be computationally expensive for large datasets. To address this challenge, utility learning methods have been introduced.

Utility learning aims to approximate the true utility function in a data-driven way. The general formulation of utility learning can be expressed as: $\hat{U}(S) = h(g(S))$ where $S$ is a subset of data points, $g(\cdot)$ is a feature extractor that captures relevant characteristics of the subset, and $h(\cdot)$ is a learnable function that maps these characteristics to an estimated utility value. This approach differs from the original utility function by avoiding direct model retraining for each subset, instead learning to predict utility based on subset features. To train this approximation function, we construct a dataset of utility samples. Each sample consists of a subset of data points and its corresponding true utility value, obtained by evaluating the original utility function on a limited number of subsets.

A notable instance of utility learning is the approach for training data valuation proposed by Wang et al. (2021). This method efficiently estimates model performance on various subsets of the training data without repeated model retraining. It represents subsets as binary vectors as $g(S)$, where each element corresponds to a specific data point in the training set, indicating its presence or absence. The method uses a regression model $\hat{U}_{\text{train}}$ to approximate the ground truth utility function $U : 2^{\mathcal{D}_{\text{tr}}} \to \mathbb{R}$, where $U(S) = \text{Acc}(f_S, \mathcal{D}_{\text{val}})$. Here, $f_S$ denotes a model trained on the subset $S \subseteq \mathcal{D}_{\text{tr}}$, and $\mathcal{D}_{\text{val}}$ is the validation set. The regression model is trained on a set of utility samples to minimize the difference between predicted and true utility values.

While this approach is effective for training data, it faces significant challenges when applied to graph inference data valuation: **1. Player Dependence:** The input of $\hat{U}_{\text{train}}$, using player-specific dummy variables, is tied to a specific set of players (data points) and cannot be directly transferred to new player sets or games. In the context of graph inference data valuation, this limitation becomes

particularly problematic. While we have access to the labels of the validation graph $\mathcal{G}_{\text{Val}}$, allowing us to compute the validation accuracy for different subsets of neighbor nodes on labeled validation nodes, the utility function learned on $S \subseteq \mathcal{N}(\mathcal{V}_{\text{Val}}^l) \in \mathcal{G}_{\text{Val}}$ cannot be directly applied to test nodes in $\mathcal{G}_{\text{Te}}$. This is because the test nodes represent a new set of players from $\mathcal{N}(\mathcal{V}_t)$, which were not present during the utility function learning process. **2. Indirect optimization:** As discussed in Section 2.1, the utility function is a crucial input for data valuation solutions. Following this logic, current utility learning approaches employ a two-stage process: learning the utility function with sampled training subsets and corresponding validation accuracy, then estimating Shapley values with the fixed learned utility. This indirect method has two main drawbacks: (a) it requires handling computationally expensive accuracy-level data, and (b) optimizing for accuracy prediction doesn't necessarily lead to optimal Shapley value estimation.

## 4.2 SHAPLEY-GUIDED UTILITY LEARNING

To address the challenges of player dependence and indirect optimization in graph inference data valuation, we propose Shapley-Guided Transferable Utility Learning (SGUL). This novel approach focuses on two key aspects: (1) Transferable Feature Extraction: We introduce a method that transforms player-dependent inputs into transferable, performance-related features. These features capture both graph structure and model behavior without relying on test labels, enabling our utility learning model to generalize across different player sets and graphs. (2) Shapley-guided Optimization: We develop a method that enables direct optimization of Shapley values, addressing the limitations of indirect optimization approaches.

### 4.2.1 TRANSFERABLE FEATURE EXTRACTION

As mentioned in Section 4.1, traditional utility learning approaches often use player-specific binary vectors as input (Wang et al., 2021). While effective for training data, this method faces limitations when applied to graph inference data valuation, particularly due to its inability to transfer to new player sets. To overcome the limitations of traditional utility learning approaches, we propose a novel feature extraction method that transforms player-dependent inputs into transferable, performance-related features. This approach builds upon the general utility learning formulation introduced in Equation (4.1), focusing on designing a transferable feature extractor $g(S)$. Specifically, our proposed $g(S)$ aims to capture both structural and model-specific characteristics of the graph data. The input $S$ is a set of neighboring nodes of target test nodes. As discussed in Section 2.2, we also employ a permutation sampling process for estimating *Structure-Aware Shapley value*. During this process, we incrementally add neighboring nodes to target test nodes following permissible permutations. This generates a series of test subgraphs $\mathcal{G}_{\text{sub}} = (\mathcal{V}_{\text{sub}}, \mathcal{E}_{\text{sub}}, \mathbf{X}_{\text{sub}})$, each including the added neighboring nodes and target test nodes. The function $g(\cdot)$ maps the current neighbor node set $S \in \mathcal{V}_{\text{sub}}$ to a $d$-dimensional feature vector $\mathbf{x} \in \mathbb{R}^d$, which encapsulates the characteristics of the subgraph induced by $S$ as derived from the permutation. These features serve as proxies for model accuracy, capturing both graph structure and GNN behavior without relying on true labels.

Our feature vector $\mathbf{x}$ comprises two main categories: data-specific and model-specific features. Data-specific features capture graph structure and test node relationships to the training set, including edge cosine similarity, representation distance, and classwise representation distance. Model-specific features assess prediction confidence and uncertainty using the GNN model's output. These include maximum predicted confidence, target class confidence, propagated confidence measures, negative entropy, and confidence gap. By combining these features, we create a comprehensive representation of GNN performance on test subgraphs without relying on true labels. This approach allows us to estimate the utility function effectively, even in the absence of ground truth information for test nodes. A detailed description and mathematical formulation of each feature is provided in Appendix A.

### 4.2.2 SGUL-ACCURACY AND ITS LEARNING PROCESS

With our transferable feature extraction approach, we can adapt traditional utility learning methods to create an accuracy-oriented variant of our method. Following (Wang et al., 2021), we can apply these features directly in a standard regression framework. This base model, termed SGUL-Accuracy, serves as an important comparison point for our proposed Shapley-guided method in Section 4.2.3. For a detailed description of SGUL-Accuracy training process, please refer to Appendix B.

### 4.2.3 SHAPLEY-GUIDED UTILITY LEARNING (SGUL)

Having established our transferable utility learning framework and selected features, we now address the challenge of efficiently optimizing our utility function. Traditional accuracy-based approaches we seen in Section 4.1 employ a two-stage optimization process: first learning a utility function to predict model accuracy, then using this function to estimate Shapley values. However, this indirect accuracy-based method has significant limitations. It is computationally expensive due to the large amount of accuracy-level data required, especially for large-scale graphs. More importantly, the optimization process in these accuracy-oriented methods is misaligned with the ultimate goal of data valuation. While the first stage focuses on accurately predicting model accuracy, it doesn't directly minimize the difference between predicted Shapley values (derived from the learned utility function) and true Shapley values (calculated using actual accuracy). Our aim is to develop a method that directly optimizes for accurate Shapley value prediction, ensuring that the learned utility function could produce Shapley values that closely match those derived from true accuracy measurements.

Through theoretical analysis of the Shapley value definition, we discover a key insight: the Shapley value calculation is a deterministic linear transformation of the utility function. This insight is formalized in the following theorem:

**Theorem 1** (Shapley Value Decomposition). *Given a linear utility function $U(S) = \mathbf{w}^\top \mathbf{x}(S)$, where $\mathbf{w} \in \mathbb{R}^d$ is a parameter vector and $\mathbf{x}(S) \in \mathbb{R}^d$ is a feature vector representing subset $S$, the Shapley value of player $i$ with respect to $U$ can be expressed as a linear combination of Feature Shapley Values:*

$$\phi_i(U) = \mathbf{w}^\top \boldsymbol{\psi}_i \qquad (2)$$

*where $\boldsymbol{\psi}_i = [\phi_i(U_1), \phi_i(U_2), \ldots, \phi_i(U_d)]^\top$ is the vector of Feature Shapley Values, and $U_k(S) = x_k(S)$ is the utility function considering only the $k$-th feature.*

The proof of this theorem is provided in Appendix C. This theorem establishes a direct link between learnable parameters and fitted Shapley values: $\hat{\phi}_i(U) = \mathbf{w}^\top \boldsymbol{\psi}_i$, where $\hat{\phi}_i(U)$ is the fitted Shapley value for $i$, $\mathbf{w}$ is our learnable parameter vector, and $\boldsymbol{\psi}_i$ is the Feature Shapley vector for $i$. Unlike previous decoupled accuracy-oriented optimization that solve $\hat{\phi}_i(\hat{U})$ subject to $\hat{U}(S) = \arg\min \mathcal{L}(\hat{U}(S), U(S))$ ($\mathcal{L}$ is the loss), our approach directly optimizes the fitted Shapley values from learned utility function with the help of this theorem.

Building on this, we propose Shapley-Guided Transferable Utility Learning (SGUL), which integrates our transferable utility learning framework with a Shapley-guided optimization method. To learn the optimal parameter vector $\mathbf{w}$, we formulate the following optimization problem:

$$\min_{\mathbf{w}} \sum_{i \in \mathcal{N}(\mathcal{V}_{\text{Val}})} (\phi_i(U) - \mathbf{w}^\top \boldsymbol{\psi}_i)^2 + \lambda \|\mathbf{w}\|_1 \qquad (3)$$

Here, $\mathcal{V}_{\text{Val}}$ represents the set of labeled validation nodes, and $\mathcal{N}(\cdot)$ denotes the set of neighbors for a given node set. The regularization parameter $\lambda$ controls the trade-off between fitting the data and model complexity. This optimization problem is designed to find the optimal weight vector $\mathbf{w}$ that quantifies the importance of each feature in the utility function. The input features are the Feature Shapley Values $\boldsymbol{\psi}_i$ for each neighbor $i$, computed as a preprocessing step. The target variable is the true Shapley value $\phi_i(U)$, which we can compute using the known accuracy on the validation set.

By solving this optimization problem, we obtain the optimal parameter vector $\mathbf{w}^*$ that minimizes the difference between predicted and true Shapley values across all validation nodes and their neighbors. This direct optimization approach contrasts with traditional methods that focus on predicting accuracy rather than Shapley values. The inclusion of $L_1$ regularization ($\|\mathbf{w}\|_1$) in the objective function promotes sparsity in the learned weights, effectively identifying the most relevant features for Shapley value estimation and helping prevent overfitting. Importantly, the learned coefficients are interpretable, as our proposed features are theoretically positively correlated with accuracy. To ensure this interpretability, we constrain the parameters to be non-negative during the learning process. Once we have obtained the optimal parameter vector $\mathbf{w}^*$, we can apply SGUL to estimate Shapley values for neighbors of test nodes. When comparing our proposed methods with different optimization protocols, we refer to the one optimized using our Shapley-guided method as SGUL-Shapley, which differs from SGUL-Accuracy, optimized for accuracy as mentioned in Subsection 4.2.2.

## 5 RELATED WORK

**Data Valuation Methods.** Data valuation approaches based on cooperative game theory, including Data Shapley (Ghorbani & Zou, 2019) and its variants (Kwon & Zou, 2021; Wang & Jia, 2023), have established effective frameworks for quantifying individual data contributions. Recent innovations include learning-agnostic frameworks (Just et al., 2023) and training-free methods (Nohyun et al., 2022). For graph-specific valuation, Winter value-based methods (Chi et al., 2024a) and task-agnostic frameworks (Falahati & Amiri, 2024) address the unique challenges of interconnected data. Data utility learning (Wang et al., 2021) enhances valuation efficiency by predicting model performance on data subsets without repeated training, improving methods like Shapley value calculations.

**Graph Neural Networks.** Graph Neural Networks (GNNs) have revolutionized graph-structured data analysis since the introduction of spectral convolutions (Bruna et al., 2013), with architectures like Graph Convolutional Networks (GCN) (Kipf & Welling, 2016) and attention-based variants (Veličković et al., 2017) gaining wide adoption. Central to GNNs' success is the message-passing mechanism (Xu et al., 2018), enabling effective information propagation across the graph. While traditionally viewed as crucial for both training and inference, recent research highlights its particular importance during testing. The introduction of PMLPs (Yang et al., 2023), which are identical to standard MLPs in training but adopt GNN's architecture with message passing in testing, reveals the critical role of testing-time structure in graph-based models' performance.

**Model Evaluation Without Labels.** Several approaches have been developed for model evaluation without test labels. Label-free model evaluation methods (see Appendix D.4) estimate model accuracy through confidence scores and distribution metrics. ATC (Garg et al., 2022) learns confidence thresholds and DoC (Guillory et al., 2021) measures confidence shifts between validation and test sets. These methods have been extended to graph domain through GNNEvaluator (Zheng et al., 2024b), which uses discrepancy attributes to train a GCN regressor for accuracy prediction. However, these methods focus on single-accuracy estimation rather than data value assessment. Recently, retraining-based approaches (see Appendix D.5) have emerged as an alternative strategy. Projection Norm (Yu et al., 2022) predicts performance by analyzing parameter changes after pseudo-label retraining, which has been adapted to graphs through LEBED (Zheng et al., 2024a). However, their computational demands make them impractical for evaluating multiple subgraph configurations required in our setting. While test-time adaptation methods (see Appendix D.6) such as GTRANS (Jin et al., 2022b) and IGT3 (Pi et al.) also operate on unlabeled test data, they focus on the one-time performance maximization through graph transformation or parameter adaptation. For the graph-inference data valuation problem requiring evaluation across numerous subgraph configurations, only non-retraining label-free model evaluation methods serve as suitable surrogate utility functions, as they enable efficient prediction across multiple subgraphs.

For a more detailed discussion on related work, please refer to Appendix D.

## 6 EXPERIMENTS

### 6.1 DATASETS AND EXPERIMENTAL SETUP

We evaluated our proposed SGUL framework on seven diverse real-world graph datasets, covering both homophily and heterophily scenarios. Our experiments focus primarily on the inductive node classification task, with additional evaluations in the transductive setting. We employed different fixed GNN models for each setting to highlight the importance of testing structures. For a detailed description of the datasets and experimental setup, please refer to Appendix E.

### 6.2 EVALUATION PROCESS AND METRICS

To evaluate our proposed SGUL framework, we design a comprehensive assessment protocol consisting of utility learning and data valuation phases. During utility learning, we follow the process detailed in Algorithm 1, which involves generating permutations on the validation graph to learn our utility function. This standardized process is applied identically across all methods to ensure fair comparison. In the data valuation phase, we apply the learned utility functions to the test graph $\mathcal{G}_{\text{Te}}$ to

predict accuracies for target nodes $\mathcal{V}_t$. These predicted accuracies serve as utility function outputs $U(\cdot)$ for calculating Shapley values following Algorithm 2.

To assess the quality of data values produced by SGUL, we employ a **node dropping experiments** that targets high-value nodes in the graph. This experiment is one of the most common and widely used evaluation methods in data valuation research (Ghorbani & Zou, 2019; Jiang et al., 2023; Kwon & Zou, 2023; Chi et al., 2024a). By sequentially removing nodes ranked according to their assessed values, we can observe a *node dropping performance curve*, which visually represents how the model's performance degrades as important nodes are removed. A good data valuation method should result in a curve that shows a rapid and sharp decline in model performance. This corresponds to identifying a crucial data subset, whose removal would significantly affect performance. Moreover, the decrease in performance caused by our method should not only be substantial but also persistent throughout the node-dropping process, ensuring that the importance identification is highly consistent (Ghorbani & Zou, 2019; Jiang et al., 2023; Chi et al., 2024a). To quantify the effectiveness of a valuation method, we use the Area Under the Curve (AUC) metric of these *node dropping performance curves*. The AUC provides a single numerical value that captures the overall behavior of the node dropping process, reflecting both the initial rapid decline and the sustained performance drop. A lower AUC indicates a better valuation method, as it represents a more rapid and sustained decline in model accuracy when high-value nodes are removed. For detailed implementation of this evaluation protocol, please refer to Algorithm 3.

## 6.3 BASELINES

To evaluate the effectiveness of our proposed SGUL framework, we compare it with several alternative utility learning methods. All these baselines aim to approximate the true utility function for graph inference data valuation, but differ in their specific techniques. The baselines we consider are: **i. Average Thresholded Confidence (ATC):** ATC (Garg et al., 2022) estimates accuracy by learning a threshold on the model's confidence scores. We implement two variants: **ATC-MC** (Maximum Confidence) and **ATC-NE** (Negative Entropy). **ii. Difference of Confidence (DoC):** DoC (Guillory et al., 2021) measures the difference in average confidence between the validation and test sets to predict accuracy change. **iii. GNNEvaluator:** GNNEvaluator (Zheng et al., 2024b) is a novel method designed to assess GNN performance on unseen graphs without labels. **iv. Natural Confidence-Based Baselines:** We also include two straightforward confidence-based baselines: **(a) Maximum Confidence**: This uses the highest confidence score across all nodes and classes in the subgraph. **(b) Class Confidence**: This calculates the average confidence score of the predicted class for each test node, using the full test graph as input. We ensure a fair comparison by using the same overall framework for all methods, including identical sampling procedures for validation and test permutations. For a complete description of those methods and corresponding training process, please see Appendix D.3 and Appendix B respectively.

## 6.4 MAIN RESULTS AND ANALYSIS

To evaluate the effectiveness of our proposed SGUL framework, we conducted comprehensive experiments on various graph datasets using both SGC and GCN models in the inductive and Transductive setting. Our analysis focuses on two key aspects: the detailed behavior of node removal across different datasets and the overall performance of SGUL compared to baselines.

### 6.4.1 INDUCTIVE NODE REMOVAL ANALYSIS

To provide a nuanced understanding of how different methods perform as nodes are progressively removed, we present accuracy curves for node dropping experiments. Figures 1 and 2 show these curves for SGC and GCN models, respectively, across various datasets. Figure 1 illustrates the accuracy curves for node dropping experiments using SGC models across various datasets. These results reveal two key characteristics of our SGUL method. Firstly, SGUL consistently demonstrates a steeper initial drop in accuracy when removing the first few nodes, particularly evident in datasets like Cora and Citeseer. This rapid initial decline indicates SGUL's superior ability to identify the most critical nodes in the graph structure. Secondly, as more nodes are removed, SGUL maintains a more stable accuracy curve compared to other methods. This long-term stability is especially pronounced in larger datasets such as CS and Physics, suggesting a more robust and consistent

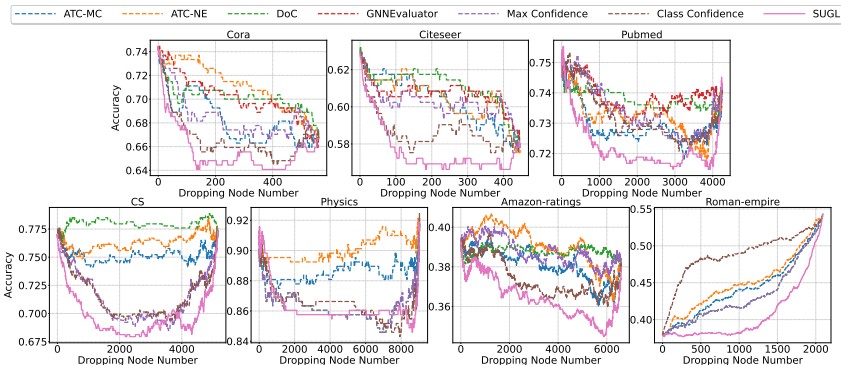

Figure 1: Accuracy curves for node dropping experiments using the SGC model on various datasets in the inductive setting. Our proposed `SGUL` method consistently maintains higher accuracy as nodes are removed, indicating its effectiveness in identifying important nodes. Note that GNNEvaluator is not shown for the larger datasets due to Out of Memory (OOM) errors.

ranking of node importance throughout the removal process. The performance of `SGUL` varies across datasets, showcasing its adaptability to different graph structures. For instance, in the Cora dataset, `SGUL` exhibits a clear advantage throughout the entire node removal process. In the Amazon-ratings dataset, which presents a more challenging heterophily scenario, `SGUL` consistently outperforms other methods, particularly in the latter stages of node removal. Similar performance patterns are observed when using GCN models, demonstrating that `SGUL`'s effectiveness is not limited to a specific GNN architecture. For detailed results of the GCN experiments, please refer to Appendix F. Additionally, to demonstrate scalability on much larger graphs, we conducted experiments on the ogbn-arxiv dataset, with results showing consistent improvements over baselines (see Appendix L).

### 6.4.2 TRANSDUCTIVE NODE REMOVAL ANALYSIS

To further validate the effectiveness of our `SGUL` framework, we extended our evaluation to the transductive setting using the Cora, Citeseer, and Pubmed datasets. This setting allows us to assess how our method performs when the entire graph structure is known during both training and inference. Our results show that `SGUL` consistently maintains higher accuracy as nodes are removed across all datasets for both SGC and GCN models. This performance is particularly notable in the Pubmed dataset, where `SGUL` shows a significant advantage over other methods throughout the node removal process. For a detailed analysis of the transductive setting experiments, including accuracy curves for node dropping experiments and AUC results, please refer to Appendix G.

### 6.5 ABLATION STUDIES

To further validate the effectiveness of our proposed Shapley-guided approach and to provide deeper insights into its performance, we conducted a series of ablation studies. These studies aim to compare our Shapley-guided method (`SGUL-Shapley`) with an accuracy-based optimization approach (`SGUL-Accuracy`) across different aspects of performance and efficiency.

### 6.5.1 IN-SAMPLE ERROR COMPARISON OF SHAPLEY AND ACCURACY OPTIMIZATION

To empirically validate our theoretical arguments from Section 4.2.3, we compared the performance of Shapley-guided optimization (`SGUL-Shapley`) against accuracy-based optimization (`SGUL-Accuracy`). We conducted in-sample Mean Squared Error (MSE) comparisons across various datasets in the inductive setting with the SGC model. Our results show that `SGUL-Shapley` consistently achieves lower MSE across all datasets, with statistically significant differences ($p < 0.05$). These findings provide strong empirical support for our theoretical analysis, demonstrating that directly optimizing for Shapley values leads to more accurate estimation of data value compared to the accuracy-based approach. For a detailed description of the experimental setup and full results, please refer to Appendix I.

### 6.5.2 DROPPING NODE COMPARISON OF OPTIMIZATION OBJECTIVES

We conducted an ablation study comparing Shapley value optimization (`SGUL-Shapley`) and accuracy-based optimization (`SGUL-Accuracy`) in node dropping experiments. Our results show that `SGUL-Shapley` generally outperforms `SGUL-Accuracy` across most datasets and models, achieving better results in 10 out of 14 dataset-model combinations. The improvement is particularly noticeable for larger and more complex datasets such as CS, Physics, and Amazon-ratings. These findings demonstrate that directly optimizing for Shapley values leads to more accurate and robust inference data valuation. For detailed experimental setup and full results, please refer to Appendix J.

### 6.5.3 EFFICIENCY ANALYSIS OF SHAPLEY AND ACCURACY OPTIMIZATION

To complement our effectiveness analysis, we also evaluated the computational efficiency of Shapley value optimization (`SGUL-Shapley`) compared to accuracy-based optimization (`SGUL-Accuracy`). This analysis helps validate the practical applicability of our method, especially for large-scale graph applications. We recorded the fitting time and memory usage for both methods under the OLS setting described in the In-Sample Error Comparison. The implementation used PyTorch with a learning rate of 0.001 for both methods. We ran each fitting process 10 times and averaged the results, with each run consisting of 1000 epochs. Table 1 summarizes our findings:

| Setting | Dataset | Shapley Optimization | | Accuracy Optimization | |
|---|---|---|---|---|---|
| | | Time (s) | Memory (MB) | Time (s) | Memory (MB) |
| Inductive | Cora | **0.63** | **16.28** | 0.66 | 29.58 |
| | Citeseer | **0.64** | **16.28** | 1.54 | 28.59 |
| | Pubmed | **0.56** | **16.49** | 0.67 | 87.60 |
| | CS | 0.52 | **16.53** | **0.50** | 44.19 |
| | Physics | **0.44** | **16.71** | 0.55 | 39.05 |
| | Amazon-ratings | **0.64** | **16.59** | 0.67 | 115.62 |
| | Roman-empire | **0.53** | **16.37** | 0.65 | 50.25 |
| Transductive | Cora | **0.59** | **16.35** | 0.75 | 65.77 |
| | Citeseer | **0.51** | **16.35** | 0.59 | 61.28 |
| | Pubmed | **0.53** | **16.81** | 0.67 | 72.22 |

Table 1: Comparison of training time and peak memory usage between Shapley-guided (`SGUL-Shapley`) and accuracy-based (`SGUL-Accuracy`) optimization approaches using the SGC model.

The results demonstrate that `SGUL-Shapley` consistently outperforms `SGUL-Accuracy` in terms of computational efficiency. `SGUL-Shapley` achieves faster training times for most datasets, with notable improvements for larger datasets like Citeseer and Pubmed. More significantly, `SGUL-Shapley` shows a substantial reduction in memory usage across all datasets, often using less than half the memory required by `SGUL-Accuracy`. This efficiency advantage is particularly pronounced for larger datasets such as Amazon-ratings, where `SGUL-Shapley` uses only about 14% of the memory consumed by `SGUL-Accuracy`.

These findings indicate that `SGUL-Shapley` not only provides more accurate node importance estimations but also offers significant computational benefits. The reduced memory footprint and faster training times make `SGUL-Shapley` particularly suitable for large-scale graph applications where resource efficiency is crucial. This combination of effectiveness and efficiency underscores the practical value of our Shapley-guided approach in real-world graph inference data valuation tasks.

## 7 CONCLUSION

This paper introduces Shapley-Guided Utility Learning (`SGUL`), a pioneering framework for valuing graph inference data without test labels. `SGUL` uniquely integrates transferable feature extraction with Shapley-guided optimization, addressing the challenges of generalization to unseen structures and computational efficiency in graph data valuation. Our comprehensive experiments across diverse datasets consistently demonstrate `SGUL`'s superiority over existing methods in both inductive and transductive settings.

ACKNOWLEDGMENTS

The research is supported by the National Science Foundation (NSF) under grant numbers NSF2406647 and NSF-2406648.

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

## A  SELECTED FEATURES FOR ACCURACY PREDICTION

When performing Graph Inference Data Valuation, we incrementally add neighboring nodes to the target test nodes following permissible permutations. This process generates a series of test subgraphs. Let $\mathcal{G}_{\text{sub}} = (\mathcal{V}_{\text{sub}}, \mathcal{E}_{\text{sub}}, \mathbf{X}_{\text{sub}})$ denote the current sub-test graph , which includes the added neighboring nodes and target test nodes. Our goal is to identify features that reflect the performance of the fixed GNN model on $\mathcal{G}_{\text{sub}}$ without relying on true labels. To this end, we propose a set of data-specific and model-specific features that correlate with GNN performance. These features are designed to capture both the structural properties of the graph and the behavior of the GNN model on the test nodes.

**Data-specific features:** Our data-specific features focus on graph homophily and distance measures:

1. **Edge Cosine Similarity** ($\bar{s}_e$): This feature measures the homophily level of the current subgraph. For graph neural networks, it is recognized that performance is strongly correlated with the graph's homophily level, where connected nodes tend to share similar characteristics (Zhu et al., 2020a; Li et al., 2024). A higher homophily level often leads to better GNN performance.

$$\bar{s}_e = \frac{1}{|\mathcal{E}_{\text{sub}}|} \sum_{(i,j) \in \mathcal{E}_{\text{sub}}} \cos(\mathbf{x}_i, \mathbf{x}_j)$$

where $\mathbf{x}_i$ and $\mathbf{x}_j$ are the feature vectors of nodes $i$ and $j$, respectively, and $\mathcal{E}\text{sub}$ is the set of edges in the current subgraph. This measure accurately captures the average cosine similarity between connected nodes, reflecting the homophily of the subgraph structure.

2. **Representation Distance** ($d_{\text{rep}}$): This feature quantifies the overall dissimilarity between test nodes and the training set, which has been shown to be related to GNN performance (Ma et al., 2021). A smaller feature distance typically indicates that the test nodes are more similar to the training data, which often correlates with better GNN performance on these test nodes.

$$d_{\text{rep}} = \frac{1}{|\mathcal{V}_t|} \sum_{v \in \mathcal{V}_t} \cos(\mathbf{h}_v, \bar{\mathbf{h}}_{\text{train}})$$

where $\mathbf{h}_v$ is the aggregated feature vector of the target test node $v$, and $\bar{\mathbf{h}}_{\text{train}}$ is the mean aggregated feature vector of all training nodes.

3. **Classwise Representation Distance** ($d_{\text{class}}$): This feature provides a more fine-grained measure of the distance between test nodes and training data, by considering class-specific prototypes. It captures how well the test nodes align with the class representations learned from the training data.

$$d_{\text{class}} = \frac{1}{|\mathcal{V}_t|} \sum_{v \in \mathcal{V}_t} \min_{c \in \mathcal{C}} \cos(\mathbf{h}_v, \bar{\mathbf{h}}_c)$$

where $\mathcal{C}$ is the set of classes, $\bar{\mathbf{h}}_c$ is the mean aggregated feature vector of training nodes in class $c$, and $\mathcal{V}_t$ is the set of target test nodes. This feature complements the overall representation distance by providing class-specific information, which can be particularly useful in multi-class classification tasks.

**Model-specific features:** Our model-specific features leverage the output of the fixed GNN model $f(\cdot)$ to compute confidence scores and uncertainty measures:

4. **Maximum Predicted Confidence** ($c_{\text{max}}$): The confidence of model predictions has been shown to be strongly correlated with model accuracy (Guo et al., 2017; Guillory et al., 2021; Garg et al., 2022). This feature captures the average maximum confidence score for target test nodes when the current subgraph is input to the fixed GNN model.

$$c_{\text{max}} = \frac{1}{|\mathcal{V}_t|} \sum_{v \in \mathcal{V}_t} \max_{y \in \mathcal{Y}} f_y(\mathcal{G}_{\text{sub}})_v$$

5. **Target Class Confidence** ($c_{\text{target}}$): To mitigate the impact of changing predictions as we add nodes to the test subgraph, we fix the predicted label for each node based on the full test graph. This feature represents the average confidence score of these fixed predicted classes for the target test nodes.

$$c_{\text{target}} = \frac{1}{|\mathcal{V}_t|} \sum_{v \in \mathcal{V}_t} f_{\hat{y}_v}(\mathcal{G}_{\text{sub}})_v$$

where $\hat{y}_v = \arg\max_{y \in \mathcal{Y}} f_y(\mathcal{G}_{\text{Te}})_v$ is the predicted class for node $v$ using the full test graph $\mathcal{G}_{\text{Te}}$.

6. **Propagated Maximum Confidence** ($c_{\text{prop\_max}}$): Label propagation (Zhu & Ghahramani, 2002) has proven effective in graph-based classification tasks. We first predict a distribution for each node in the current subgraph using only their features, then propagate these distributions over the graph structure.

$$c_{\text{prop\_max}} = \frac{1}{|\mathcal{V}_t|} \sum_{v \in \mathcal{V}_t} \max_{y \in \mathcal{Y}} \tilde{f}_y(\mathcal{G}_{\text{sub}})_v$$

where $\tilde{f}_y(\mathcal{G}_{\text{sub}})_v = \text{LP}(f_y(\mathcal{G}_{\text{sub}}^{\emptyset}))_v$, $\mathcal{G}_{\text{sub}}^{\emptyset} = (\mathcal{V}_{\text{sub}}, \emptyset, \mathbf{X}_{\text{sub}})$ is the subgraph with empty edge set, $f_y(\mathcal{G}_{\text{sub}}^{\emptyset})_v$ is the initial prediction for node $v$ using only its features, and $\text{LP}(\cdot)$ denotes the label propagation operation on $\mathcal{G}_{\text{sub}}$.

7. **Propagated Target Confidence** ($c_{\text{prop\_target}}$): This feature captures the average confidence of the fixed predicted class after label propagation, using the same process as in Propagated Maximum Confidence.

$$c_{\text{prop\_target}} = \frac{1}{|\mathcal{V}_t|} \sum_{v \in \mathcal{V}_t} \tilde{f}_{\hat{y}_v}(\mathcal{G}_{\text{sub}})_v$$

where $\hat{y}_v$ is the predicted class for node $v$ using the full test graph, as defined earlier, and $\tilde{f}_{\hat{y}_v}(\mathcal{G}_{\text{sub}})_v$ is the propagated probability for class $\hat{y}_v$ of node $v$.

8. **Negative Entropy** ($H_{\text{neg}}$): Recent research has shown that negative entropy effectively measures uncertainty in machine learning models and is highly correlated with accuracy (Sensoy et al., 2018; Garg et al., 2022). Higher negative entropy indicates more certain predictions.

$$H_{\text{neg}} = -\frac{1}{|\mathcal{V}_t|} \sum_{v \in \mathcal{V}_t} \sum_{y \in \mathcal{Y}} f_y(\mathcal{G}_{\text{sub}})_v \log f_y(\mathcal{G}_{\text{sub}})_v$$

9. **Confidence Gap** ($\Delta c$): This feature measures the average difference between the highest and second-highest confidence scores for the target test nodes. It can be viewed as another uncertainty measure, where a larger confidence gap suggests more certain predictions and potentially higher accuracy.

$$\Delta c = \frac{1}{|\mathcal{V}_t|} \sum_{v \in \mathcal{V}_t} \left( \max_{y \in \mathcal{Y}} f_y(\mathcal{G}_{\text{sub}})_v - \max_{y' \in \mathcal{Y} \setminus \{\hat{y}_v\}} f_{y'}(\mathcal{G}_{\text{sub}})_v \right)$$

where $\hat{y}_v = \arg\max_{y \in \mathcal{Y}} f_y(\mathcal{G}_{\text{sub}})_v$ is the predicted class for node $v$.

These model-specific features, combined with the previously described data-specific features, provide a comprehensive representation of the GNN's performance on the sub-test graph $\mathcal{G}_{\text{sub}}$ without relying on true labels. By capturing various aspects of model confidence and uncertainty, these features serve as effective proxies for model accuracy in our utility learning framework.

## B  SGUL–ACCURACY AND BASELINE MODEL TRAINING PROCESS

Our proposed transferable feature extraction method SGUL-Accuracy addresses the challenge of lacking test labels by providing a set of proxy features that capture both graph structure and model behavior. To implement this approach in the context of utility learning for graph inference data valuation, we follow a general procedure:

1. **Generate permutations:** We create a set of permutations on the validation graph $\mathcal{G}_{\text{Val}}$, following the structure-aware Shapley value formulation (Equation (1)). Each permutation results in a series of subgraphs as neighboring nodes are incrementally added.

2. **Extract features:** For each subgraph, we apply our feature extractor $g(S)$ to obtain the transferable, performance-related features $\mathbf{x} \in \mathbb{R}^d$.

3. **Obtain ground truth utilities:** We input these validation subgraphs into the fixed GNN model $f(\cdot)$ to obtain ground truth accuracy scores. These scores serve as the target values for our utility learning method.

4. **Learn utility function:** Traditional approaches typically use the feature-utility pairs obtained from steps 2 and 3 to train a model that minimizes the difference between predicted and true utility values (or accuracies). This is often formulated as a regression problem:

$$\min_{\theta} \sum_{S} (U(S) - h_{\theta}(g(S)))^2$$

where $U(S)$ is the ground truth utility (accuracy) for subgraph $S$, $g(S)$ is our feature extractor, and $h_{\theta}$ is a learnable function parameterized by $\theta$.

This procedure allows us to create a dataset of feature-utility pairs, which can be used to learn a utility function that generalizes to unseen test structures. It's worth noting that this general framework can accommodate various transferable feature extraction methods. While we have proposed a specific design for our feature extractor $g(S)$, other approaches could potentially be integrated into this framework, as long as they produce transferable, performance-related features $\mathbf{x} \in \mathbb{R}^d$.

Interestingly, this training procedure can be identically applied to label-free model evaluation methods (Garg et al., 2022; Guillory et al., 2021; Zheng et al., 2024b) which serve as baselines in our experimental part. These methods aim to assess model performance on unlabeled data by leveraging various proxy metrics or transferable features. The key difference lies in the ultimate goal: while label-free model evaluation methods focus on estimating overall model performance, our approach extends this concept to the more granular task of valuing individual graph elements for inference.

## C    PROOF OF SHAPLEY VALUE DECOMPOSITION THEOREM

Here we provide the proof for the Shapley Value Decomposition Theorem stated in Section 4.2.3.

**Theorem 2** (Shapley Value Decomposition). *Given a linear utility function $U(S) = \mathbf{w}^{\top}\mathbf{x}(S)$, where $\mathbf{w} \in \mathbb{R}^d$ is a parameter vector and $\mathbf{x}(S) \in \mathbb{R}^d$ is a feature vector representing subset $S$, the Shapley value of player $i$ with respect to $U$ can be expressed as a linear combination of Feature Shapley Values:*

$$\phi_i(U) = \mathbf{w}^{\top}\boldsymbol{\psi}_i$$

*where $\boldsymbol{\psi}_i = [\phi_i(U_1), \phi_i(U_2), \ldots, \phi_i(U_d)]^{\top}$ is the vector of Feature Shapley Values, and $U_k(S) = x_k(S)$ is the utility function considering only the $k$-th feature.*

*Proof.* We begin by recalling the definition of the Shapley value for a player $i$ with respect to a utility function $U$:

$$\phi_i(U) = \sum_{S \subseteq \mathcal{N} \setminus \{i\}} \frac{|S|!(n - |S| - 1)!}{n!} [U(S \cup \{i\}) - U(S)]$$

where $\mathcal{N}$ is the set of all players and $n = |\mathcal{N}|$.

Given the linear utility function $U(S) = \mathbf{w}^{\top}\mathbf{x}(S)$, we can substitute this into the Shapley value definition:

$$\phi_i(U) = \sum_{S \subseteq \mathcal{N} \setminus \{i\}} \frac{|S|!(n - |S| - 1)!}{n!} [\mathbf{w}^\top \mathbf{x}(S \cup \{i\}) - \mathbf{w}^\top \mathbf{x}(S)]$$

$$= \sum_{S \subseteq \mathcal{N} \setminus \{i\}} \frac{|S|!(n - |S| - 1)!}{n!} \mathbf{w}^\top [\mathbf{x}(S \cup \{i\}) - \mathbf{x}(S)]$$

$$= \mathbf{w}^\top \sum_{S \subseteq \mathcal{N} \setminus \{i\}} \frac{|S|!(n - |S| - 1)!}{n!} [\mathbf{x}(S \cup \{i\}) - \mathbf{x}(S)]$$

Now, let's consider the $k$-th component of the feature vector $\mathbf{x}(S)$, which we denote as $x_k(S)$. We can define a utility function $U_k(S) = x_k(S)$ that considers only this $k$-th feature. The Shapley value for player $i$ with respect to $U_k$ is:

$$\phi_i(U_k) = \sum_{S \subseteq \mathcal{N} \setminus \{i\}} \frac{|S|!(n - |S| - 1)!}{n!} [x_k(S \cup \{i\}) - x_k(S)]$$

Comparing this with the last line of our previous derivation, we can see that:

$$\phi_i(U) = \mathbf{w}^\top [\phi_i(U_1), \phi_i(U_2), \dots, \phi_i(U_d)]^\top = \mathbf{w}^\top \boldsymbol{\psi}_i$$

where $\boldsymbol{\psi}_i = [\phi_i(U_1), \phi_i(U_2), \dots, \phi_i(U_d)]^\top$ is the vector of Feature Shapley Values.

This completes the proof of the Shapley Value Decomposition Theorem. $\square$

This theorem demonstrates that for a linear utility function, the Shapley value can be decomposed into a linear combination of Feature Shapley Values derived from Appendix A. This decomposition forms the theoretical foundation for our Shapley-Guided Generalizable Utility Learning (SGUL) method, allowing us to efficiently learn utility function parameters by optimizing for Shapley values directly.

## D   EXTENDED RELATED WORK

### D.1   DATA-EFFICIENT LEARNING ON GRAPHS

Data-efficient learning on graphs primarily uses two approaches to address limited labeled data. Graph self-supervised learning develops representations without labels, where contrastive learning methods like Deep Graph Infomax (Velickovic et al., 2019) and GRACE (Zhu et al., 2020b) contrast different graph views, while non-contrastive methods such as BGRL (Thakoor et al., 2021) and Graph Barlow Twins (Bielak et al., 2022) achieve strong performance without negative samples. Recent works address sampling bias (Zhao et al., 2021; Xia et al., 2022) and develop feature augmentation techniques Zhang et al. (2022; 2023). Chi & Ma (2024) enhance contrastive learning through node similarity while Ma et al. (Ma et al., 2024) establish comprehensive benchmarks for evaluation. Meanwhile, graph active learning optimizes node selection for labeling, with methods like AGE (Cai et al., 2017) combining multiple selection metrics, GPA (Hu et al., 2020a) using sequential decision-making, and GRAIN (Zhang et al., 2021c)/RIM (Zhang et al., 2021b) reformulating selection as influence maximization. Advanced techniques include LSCALE (Liu et al., 2022) exploiting labeled and unlabeled representations, ALG (Zhang et al., 2021a) considering both representativeness and informativeness, and GALclean (Chi et al., 2024b) addressing active learning for graphs with noisy structures. Recent works have also explored uncertainty quantification on graphs, with JuryGCN (Kang et al., 2022) providing deterministic uncertainty estimates through jackknife confidence intervals and Fuchsgruber et al. (2024) establishing principled approaches to uncertainty sampling for active learning on graphs. Another promising direction is graph condensation, which aims to distill large graphs into smaller synthetic versions that preserve training performance. Jin et al. (2021) introduce this problem by matching GNN training trajectories through gradient matching, while their follow-up work (Jin et al., 2022a) accelerates the process with one-step gradient matching. Recent advances by Gong et al. (Gong et al., 2025) address scalability challenges for evolving graph data through class-wise

clustering on aggregated features, achieving significant speedups while maintaining comparable performance. These data-efficient approaches complement graph inference data valuation, extending efficiency principles from training to the test-time inference phase.

## D.2 GRAPH TRAINING DATA VALUATION

While Data Shapley and subsequent methods (Ghorbani & Zou, 2019; Kwon & Zou, 2021; Wang & Jia, 2023) have been effective for i.i.d. data, they face significant challenges when applied to graph-structured data. A key challenge is capturing the hierarchical and dependent relationships among graph elements. In Graph Neural Networks (GNNs), a node's contribution to model performance is intricately linked to its position within the computation tree and its relationships with other nodes. Traditional Shapley value calculations, which treat all players (nodes) independently, fail to account for these crucial dependencies, potentially leading to inaccurate valuations.

To address these challenges, Chi et al. (2024a) proposed a more granular approach to graph data valuation. This approach considers individual nodes within the computation tree as the basic units for valuation, allowing for a more nuanced assessment of each element's contribution to GNN performance. Building upon the Shapley value formulation, this method introduces two key constraints to the set of permutations $\Pi(\mathcal{D})$:

**Level Constraint**: This constraint ensures that nodes within the same subtree of the computation graph are grouped together in the permutation. Formally, for a node $v$ in the computation tree and its descendants $\mathcal{D}(v)$, the constraint can be expressed as:

$$|\pi[i] - \pi[j]| \leq |\mathcal{D}(v)|, \quad \forall i, j \in \mathcal{D}(v) \cup \{v\}$$

where $\pi[i]$ denotes the position of node $i$ in permutation $\pi$. This preserves the hierarchical structure of the computational graph and prevents evaluation bias that could occur when players from the same group are placed at widely separated positions in the permutation.

**Precedence Constraint**: This constraint guarantees that a node appears in the permutation only after its ancestors. For a node $v$ and its ancestor set $\mathcal{A}(v)$, the constraint can be formulated as:

$$\pi[a] < \pi[v], \quad \forall a \in \mathcal{A}(v)$$

This reflects the dependency structure in GNNs, where a node's contribution is contingent on the presence of its ancestors in the computation tree.

These constraints allow for a more accurate valuation of graph data by respecting the inherent structure and dependencies within GNNs.

### D.2.1 DISCUSSION ON THE DIFFERENCE BETWEEN PC-WINTER AND SGUL

While PC-Winter pioneered the exploration of graph data valuation by introducing constraints to capture hierarchical dependencies, our work focuses specifically on the challenging scenario of test-time graph inference valuation, where ground truth labels are unavailable. Specifically, PC-Winter addresses training data valuation by defining hierarchical elements within computation trees as the data valuation objects (players), applying both Level and Precedence Constraints to capture structural dependencies. In contrast, our work (Section 4.2.3) focuses on quantifying the importance of neighbors for test nodes during inference time. We adopt the Precedence Constraint from PC-Winter while omitting the Level Constraint, as explained in Section 3.2. This design choice reflects the distinct nature of test-time neighbor relationships, which lack the clear hierarchical groupings present in training data computation trees. The Precedence Constraint proves valuable in capturing the dependencies between nodes in the message-passing process during inference.

A key technical distinction lies in our approach to utility function design. While PC-Winter leverages validation accuracy as their utility measure, the absence of test labels in our setting necessitates a novel solution. As detailed in Section 4, we introduce transferable data-specific and model-specific features that can effectively approximate model performance without ground truth labels. This innovation enables the evaluation of neighbor importance during inference time.

Our work complements PC-Winter by extending graph data valuation to test-time scenarios, particularly crucial for applications like real-time recommendation systems and dynamic graphs where

Table 2: Comparison between PC-Winter and our Structure-aware Shapley Value with `SGUL`

| Aspect | PC-Winter | Structure-aware Shapley with `SGUL` |
|---|---|---|
| **Valuation Target** | Training graph elements | Test-time neighbors |
| **Constraints Used** | Level and Precedence Constraints | Precedence only |
| **Primary Challenge** | Hierarchical dependencies | No test labels |
| **Utility Function** | Validation accuracy | Learned test accuracy |

test-time structure evaluation is essential. Table 2 highlights the key differences between our work and PC-Winter:

In our experimental analysis, we further demonstrate the effectiveness of our approach in identifying influential test-time graph structures. Through comprehensive node-dropping experiments (Section 6), we show that our method consistently outperforms baselines in terms of Area Under the Curve (AUC) scores and accuracy curve characteristics across various datasets and model architectures. These results validate the practical value of our test-time graph inference valuation framework, complementing the contributions of PC-Winter in the training data valuation setting.

### D.3 LABEL FREE MODEL EVALUATION BASELINES

In this section, we provide detailed descriptions of the baselines used for comparison in our study. These baselines represent state-of-the-art methods for label-free model evaluation, which is crucial for assessing model performance on unseen data without access to ground truth labels. The baselines we consider are:

#### D.3.1 AVERAGE THRESHOLDED CONFIDENCE (ATC)

ATC, introduced by Garg et al. (2022), estimates accuracy by learning a threshold on the model's confidence scores. We implement two variants:

(a) **ATC-MC (Maximum Confidence):**

$$\text{ATC-MC} = \frac{1}{|\mathcal{V}_{\text{test}}|} \sum_{v \in \mathcal{V}_{\text{test}}} \mathbb{I}\left[\max_{c \in \mathcal{C}} f_\theta(G_{\text{test}})_{v,c} > t\right]$$

This equation counts the fraction of test nodes where the maximum confidence exceeds a threshold $t$. Here, $f_\theta(G_{\text{test}})_{v,c}$ is the confidence score for class $c$ on test node $v$, and $t$ is determined using the validation set.

(b) **ATC-NE (Negative Entropy):**

$$\text{ATC-NE} = \frac{1}{|\mathcal{V}_{\text{test}}|} \sum_{v \in \mathcal{V}_{\text{test}}} \mathbb{I}\left[-\sum_{c \in \mathcal{C}} f_\theta(G_{\text{test}})_{v,c} \log f_\theta(G_{\text{test}})_{v,c} > t\right]$$

This variant uses negative entropy of predicted probabilities as the confidence measure, counting nodes where it exceeds the threshold.

#### D.3.2 DIFFERENCE OF CONFIDENCE (DoC)

DoC, proposed by Guillory et al. (2021), measures the difference in average confidence between the validation and test sets to predict accuracy change:

$$\text{DoC} = \text{Acc}(f, \mathcal{G}_{\text{Val}}) + \beta \cdot (\bar{c}_{\text{Te}} - \bar{c}_{\text{Val}})$$

where $\bar{c}_{\text{Val}} = \frac{1}{|\mathcal{V}_{\text{Val}}|} \sum_{v \in \mathcal{V}_{\text{Val}}} \max_{c \in \mathcal{C}} f(v)_c$ and $\bar{c}_{\text{Te}} = \frac{1}{|\mathcal{V}_{\text{Te}}|} \sum_{v \in \mathcal{V}_{\text{Te}}} \max_{c \in \mathcal{C}} f(v)_c$ are the average maximum confidences on validation and test sets respectively, $\text{Acc}(f, \mathcal{G}_{\text{Val}})$ is the accuracy on the validation graph, and $\beta$ is learned through linear regression on the validation set.

### D.3.3 GNNEVALUATOR

GNNEvaluator, introduced by Zheng et al. (2024b), is designed to assess GNN performance on unseen graphs without labels. It employs a two-stage approach:

1. It constructs a DiscGraph set, leveraging validation subgraphs from sample permutations as meta-graphs. For each meta-graph, it computes discrepancy attributes by comparing GNN embeddings and predictions between the meta-graph and original training graph.

2. A two-layer GCN regressor is trained on the DiscGraph set to estimate node classification accuracy. The regressor learns to map the discrepancy attributes to expected model performance.

For inference, GNNEvaluator computes discrepancy attributes for the unseen test graph using the fixed pre-trained GNN, then applies the trained regressor to estimate accuracy without requiring labels.

### D.3.4 NATURAL CONFIDENCE-BASED BASELINES

We also include two straightforward confidence-based baselines:

(a) **Maximum Confidence**: This uses the highest confidence score across all nodes and classes in the subgraph.

(b) **Class Confidence**: This calculates the average confidence score of the predicted class for each test node, using the full test graph as input.

We ensure a fair comparison by using the same overall framework for all methods, including identical sampling procedures for validation and test permutations.

### D.4 BASELINE METHODS AND THEIR DISTINCTIONS TO SGUL

Recent advances in label-free model evaluation have produced several notable approaches. Average Thresholded Confidence (ATC) (Garg et al., 2022) estimates model accuracy by learning appropriate thresholds on confidence scores. The method operates by computing the fraction of examples where model confidence exceeds a learned threshold, with variants utilizing either maximum confidence (ATC-MC) or negative entropy (ATC-NE) as the underlying metric. Difference of Confidence (DoC) (Guillory et al., 2021) takes a comparative approach, measuring the discrepancy in average confidence between validation and test sets to predict accuracy changes. This method leverages the observation that shifts in model confidence often correlate with performance degradation. GNNEvaluator (Zheng et al., 2024b) introduces a more sophisticated framework specifically designed for graph neural networks, employing a two-stage approach that first constructs a DiscGraph set to capture distribution discrepancies and then trains a GCN regressor to estimate node classification accuracy without requiring labels.

While these methods represent significant advances in label-free model evaluation area, they fundamentally differ from our graph inference data valuation framework in both objectives and operational mechanisms. The primary distinction lies in the granularity and scope of evaluation. Traditional label-free methods focus on estimating overall model performance on a fixed test graph, essentially treating the evaluation as a single-point estimation problem. In contrast, SGUL performs fine-grained analysis by evaluating numerous subgraph configurations to quantify each structural component contribution to model performance. This decomposition-based approach enables us to understand not just how well a model performs, but also which graph structures are crucial for that performance. The computational demands of our approach also create unique challenges - while methods like GNNEvaluator work well for one-time evaluation, they become computationally prohibitive when applied to the many subgraph permutations required for Shapley value computation, often encountering memory limitations on medium-sized datasets. Our SGUL framework addresses these challenges through specialized optimization techniques and efficient feature extraction methods that enable scalable evaluation across multiple subgraph configurations. Furthermore, the architecture of existing methods is not optimized for repeated utility assessment across permutations, as they were designed for single-pass evaluation rather than the iterative valuation process required for computing structure-aware Shapley values. Through our experimental validation, we demonstrate that SGUL not only provides more detailed structural insights but also achieves superior accuracy in estimating the

importance of individual graph components, outperforming these adapted baseline methods in the specific context of graph inference data valuation.

### D.5 RETRAINING-BASED LABEL-FREE MODEL EVALUATION METHODS

Notably, recent label-free model evaluation researches for predicting model test performance have explored retraining-based approaches, which offer novel perspectives but face limitations in our graph inference data valuation context. Projection Norm (Yu et al., 2022) predicts out-of-distribution performance by analyzing model parameter changes after retraining with pseudo-labels generated from test samples. The method demonstrates strong theoretical guarantees for linear models by measuring the distance between original and retrained model parameters as an indicator of distribution shift. Simultaneously, LEBED (Zheng et al., 2024a) approaches test-time graph distribution shifts by quantifying discrepancies in learning behaviors between training and test graphs through a GNN retraining strategy with parameter-free optimality criteria that capture both node prediction and structure reconstruction aspects.

While these methods present innovative solutions for general test accuracy prediction, they encounter fundamental constraints in our graph inference data valuation framework. The primary challenge stems from our need to evaluate $|\Omega(N(V_t))| \times |N(V_t)|$ different subgraphs for computing structure-aware Shapley values. Each subgraph configuration would demand a separate retraining process under these approaches, rendering them computationally infeasible at scale. Moreover, methods like LEBED require access to the original training graph for comparative analysis - an assumption that may not hold in practical deployment scenarios where training data access is restricted due to privacy concerns. These inherent limitations motivate our adoption of more computationally efficient approaches that can evaluate subgraph utilities without model retraining, as detailed in Section D.3.

### D.6 TEST-TIME TRAINING AND AUGMENTATION METHODS

Recent works have explored various test-time adaptation techniques for graph neural networks to address distribution shifts and OOD generalization. These methods can be broadly categorized based on their core mechanisms and objectives: GTRANS (Jin et al., 2022b) takes a data-centric approach by performing test-time graph transformation through unsupervised feature and structure augmentation. While also focusing on data adaptation, GTRANS aims to find a single optimal modified test graph that maximizes model performance, rather than evaluating the importance of individual elements. This fundamentally differs from our objective of quantifying each node's marginal contribution through structure-aware Shapley values. GOODAT (Wang et al., 2024) addresses test-time graph OOD detection as a binary classification problem by determining whether subgraphs align with the training distribution. The method employs a graph masker to compress informative subgraphs for distinguishing ID and OOD samples. However, GOODAT focuses solely on detecting OOD graphs without considering the contributions of individual nodes to model performance. IGT3 (Pi et al.) proposes a two-stage training paradigm that combines test-time training with invariant graph learning to improve OOD generalization. The method adapts model parameters through multi-level graph contrastive learning while preserving graph structure information. In contrast to our data-centric valuation framework, IGT3 modifies the model itself to adapt to distribution shifts.

While these methods present innovative solutions for improving test-time performance and OOD generalization, they differ fundamentally from our graph inference data valuation framework in both objectives and mechanisms. Our framework maintains fixed model parameters while systematically evaluating the contribution of individual nodes, enabling flexible value-based data selection for various downstream applications from performance maximization to denoising. This generality and composability of data values distinguish our approach from methods focused solely on improving overall model performance through data or model adaptation.

## E DETAILED DATASETS AND EXPERIMENTAL SETUP

### E.1 DATASETS

To evaluate the effectiveness of our proposed Shapley-Guided Utility Learning (SGUL) framework, we conducted extensive experiments on seven diverse real-world graph datasets. These datasets

include Cora, Citeseer, Pubmed (Sen et al., 2008), Coauthor-CS, Coauthor-Physics (Shchur et al., 2018), Roman-empire, and Amazon-ratings (Platonov et al., 2023). Our dataset provides excellent coverage of both homophily and heterophily in graphs. The first four datasets are characterized by homophily, where connected nodes tend to share similar features or labels. In contrast, Roman-empire and Amazon-ratings exhibit heterophily, presenting a more challenging scenario where connected nodes often have different characteristics.

### E.2 Experimental Setup

Our experiments primarily focus on the inductive node classification task, which more closely resembles real-world applications where models must generalize to unseen graph structures (Hamilton et al., 2017; Van Belle et al., 2022). In this setting, we partition each dataset into three distinct graph structures: Training Graph, Validation Graph, and Testing Graph.

To provide a comprehensive evaluation, we also extend our experiments to the transductive setting, where the entire graph structure is known during training, but only a subset of training and validation nodes has labeled data. This allows us to compare the performance of our method across different learning paradigms.

To better highlight the importance of testing structures, we employ different fixed GNN models for inductive and transductive settings:

- **Inductive Setting:** We utilize Parameterized MLPs (PMLPs) (Yang et al., 2023), which train as standard MLPs but adopt GNN-like message passing during inference. This approach emphasizes the crucial role of testing-time graph structures in model performance. During inference, PMLPs employ either Simplified Graph Convolutions (SGCs) or Graph Convolutional Networks (GCNs) (Kipf & Welling, 2016) (Wu et al., 2019) for message passing, allowing us to evaluate the impact of different aggregation schemes on our graph inference data valuation task.

- **Transductive Setting:** We use full Graph Neural Network (GNN) models, specifically GCNs and SGCs, leveraging the complete graph structure available during both training and inference. This setting allows us to assess our method's performance when the entire graph topology is known and utilized throughout the learning process.

## F GCN Model Inductive Setting Results

In addition to the SGC model results presented in the main text, we also conducted experiments using Graph Convolutional Network (GCN) models. Figure 2 shows the accuracy curves for node dropping experiments using GCN models across various datasets. The results for GCN models closely mirror those observed for SGC models. SGUL consistently outperforms other methods across various datasets, demonstrating both a steeper initial accuracy drop and better long-term stability. This consistency across different GNN architectures further validates the robustness and wide applicability of our Shapley-Guided Utility Learning framework in graph inference data valuation tasks.

## G Transductive Setting Result Analysis

To provide a comprehensive evaluation of our SGUL framework, we extended our experiments to the transductive setting using the Cora, Citeseer, and Pubmed datasets. In this setting, the entire graph structure is known during both training and inference, allowing us to assess our method's performance when the full graph topology is utilized throughout the learning process.

### G.1 Experimental Setup

For the transductive setting, we used full Graph Neural Network (GNN) models, specifically Graph Convolutional Networks (GCNs) and Simplified Graph Convolutions (SGCs). These models leverage the complete graph structure available during both training and inference.

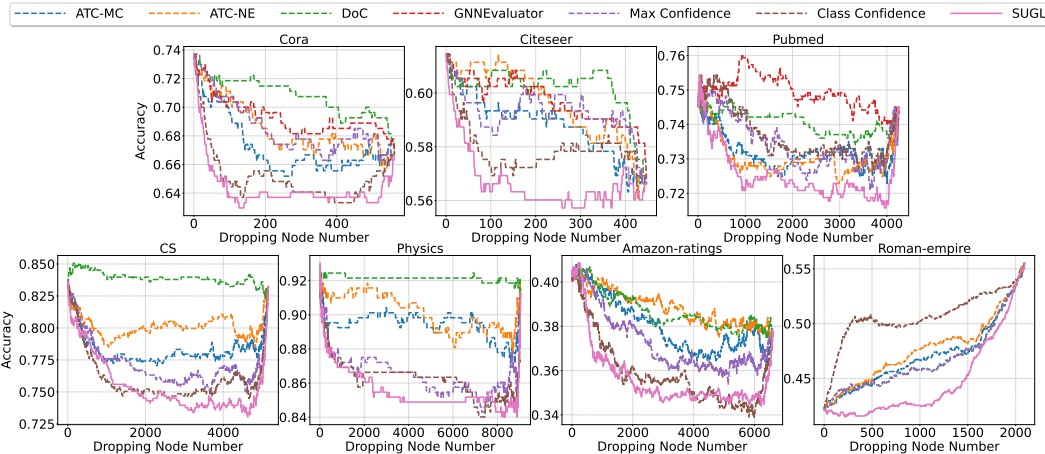

Figure 2: Accuracy curves for node dropping experiments using the GCN model on various datasets in the inductive setting. Similar to the SGC results, our proposed `SGUL` method demonstrates superior performance in maintaining higher accuracy as nodes are removed. Note that GNNEvaluator is not shown for the larger datasets due to Out of Memory (OOM) errors.

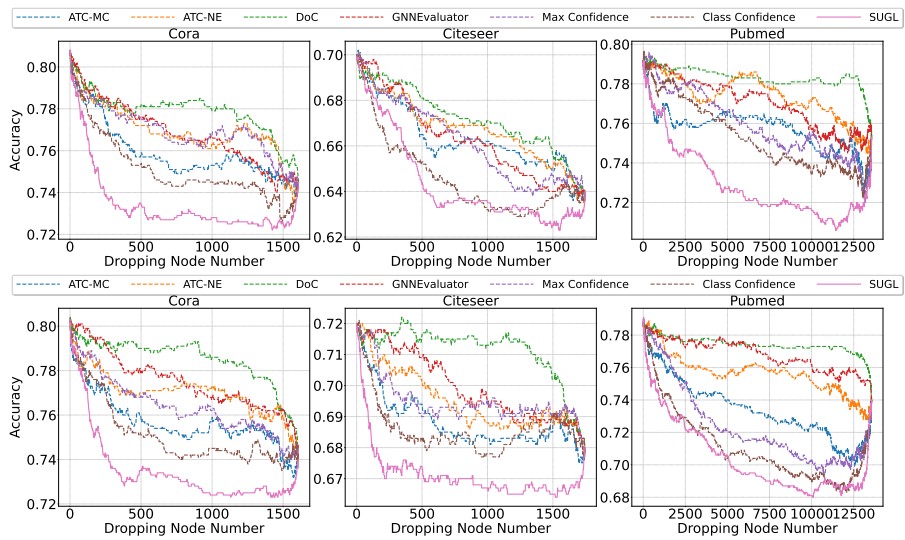

Figure 3: Accuracy curves for node dropping experiments using the SGC (above) and GCN models (below) in the transductive setting.

## G.2 NODE DROPPING EXPERIMENTS

We conducted node dropping experiments similar to those in the inductive setting. Figures 3 show the accuracy curves for node dropping experiments using SGC and GCN models respectively in the transductive setting.

According to these figures, our `SGUL` method consistently maintains higher accuracy as nodes are removed across all datasets for both SGC and GCN models. This performance is particularly notable in the Pubmed dataset, where `SGUL` shows a significant advantage over other methods throughout the node removal process.

For a more detailed quantitative analysis, including Area Under the Curve (AUC) results for both inductive and transductive settings, please refer to Appendix H.

# H    Detailed Quantitative Analysis

To provide a more comprehensive evaluation of our `SGUL` method, we present here a detailed quantitative analysis using the Area Under the Curve (AUC) metric for the node dropping process. A lower AUC score indicates superior performance, representing a more rapid decline in model accuracy when high-value nodes are removed.

## H.1    Inductive Setting Results

Table 3 presents the AUC results for node dropping experiments in the inductive setting, comparing our `SGUL` method with several baselines across different datasets and models.

Table 3: AUC results for node dropping experiments in the inductive setting. Our proposed Shap Lasso method consistently delivers competitive performance across various datasets and models. OOM refers to Out of Memory error.

| Dataset | Model | ATC-MC | ATC-NE | DoC | GNNEvaluator | Max Confidence | Class Confidence | SGUL |
|---------|-------|--------|--------|-----|--------------|----------------|------------------|------|
| **Cora** | SGC | 382.87 | 399.23 | 393.73 | 393.83 | 384.35 | 374.22 | **368.46** |
| | GCN | 377.29 | 384.54 | 397.35 | 389.21 | 384.83 | 365.48 | **361.19** |
| **Citeseer** | SGC | 269.84 | 270.44 | 273.14 | 271.21 | 269.17 | 262.43 | **257.55** |
| | GCN | 262.94 | 266.65 | 269.02 | 267.00 | 265.05 | 258.69 | **253.29** |
| **Pubmed** | SGC | 3098.02 | 3112.55 | 3136.04 | 3139.11 | 3118.71 | 3111.39 | **3068.07** |
| | GCN | 3109.80 | 3104.37 | 3147.28 | 3184.00 | 3121.70 | 3131.80 | **3080.99** |
| **CS** | SGC | 3879.30 | 3937.35 | 4023.95 | OOM | 3690.63 | 3694.78 | **3605.92** |
| | GCN | 4041.18 | 4134.89 | 4329.54 | OOM | 3984.78 | 3926.17 | **3896.88** |
| **Physics** | SGC | 8007.06 | 8138.48 | 7802.27 | OOM | 7802.27 | 7817.30 | **7797.74** |
| | GCN | 8085.59 | 8149.07 | 8328.46 | OOM | 7818.78 | 7800.91 | **7735.92** |
| **Amazon-ratings** | SGC | 2513.36 | 2577.81 | 2557.59 | OOM | 2557.58 | 2466.70 | **2401.77** |
| | GCN | 2508.83 | 2562.55 | 2551.39 | OOM | 2458.66 | 2368.02 | **2352.09** |
| **Roman-empire** | SGC | 927.38 | 944.23 | 907.65 | OOM | 907.65 | 1023.01 | **851.98** |
| | GCN | 985.70 | 997.63 | 978.04 | OOM | 978.04 | 1061.69 | **935.10** |

As evident from Table 3, our proposed `SGUL` method consistently achieves competitive performance across different datasets and models. In many cases, it outperforms the baseline methods, particularly for datasets with heterophily such as Amazon-ratings and Roman-empire. This quantitative comparison aligns with our observations from the accuracy curves, further demonstrating the effectiveness of our Shapley-Guided Utility Learning approach in capturing the importance of nodes in graph structures, even in challenging scenarios where connected nodes may have different characteristics. The superior performance of `SGUL`, as reflected in both the accuracy curves and AUC scores, can be attributed to its ability to capture the complex interactions and dependencies among nodes in the graph, leveraging the Shapley value concept to assign importance scores. By incorporating both graph structure and node features, `SGUL` provides a more comprehensive and accurate assessment of node importance compared to the baselines, which rely on simpler metrics such as confidence scores or average differences.

## H.2    Transductive Setting Results

Table 4 presents the AUC results for node dropping experiments in the transductive setting.

In the transductive setting, `SGUL` consistently outperforms all baseline methods across all datasets and models. The performance gap is even more pronounced compared to the inductive setting, particularly for larger datasets like Pubmed. These results further corroborate the effectiveness of our approach across different experimental settings.

Table 4: AUC results for node dropping experiments in the transductive setting

| Dataset | Model | ATC-MC | ATC-NE | DoC | GNNEvaluator | Max Confidence | Class Confidence | `SGUL` |
|---------|-------|--------|--------|-----|--------------|----------------|------------------|--------|
| Cora | SGC | 1219.90 | 1237.21 | 1251.55 | 1235.32 | 1237.00 | 1207.45 | **1180.11** |
|  | GCN | 1216.45 | 1238.17 | 1261.80 | 1246.09 | 1228.15 | 1207.03 | **1180.95** |
| Citeseer | SGC | 1157.87 | 1165.03 | 1172.98 | 1157.19 | 1155.54 | 1127.22 | **1123.12** |
|  | GCN | 1201.37 | 1213.33 | 1241.40 | 1221.27 | 1213.03 | 1197.15 | **1171.49** |
| Pubmed | SGC | 10271.50 | 10495.00 | 10622.11 | 10450.98 | 10302.39 | 10217.91 | **9894.75** |
|  | GCN | 9959.40 | 10262.38 | 10500.21 | 10417.29 | 9814.49 | 9648.27 | **9547.61** |

# I  IN-SAMPLE ERROR COMPARISON OF SHAPLEY AND ACCURACY OPTIMIZATION

To empirically validate our theoretical arguments from Section 4.2.3, we compared the performance of Shapley-guided optimization (`SGUL-Shapley`) against accuracy-based optimization (`SGUL-Accuracy`). We used identical validation permutation data for both methods, as described in Section 2.2. To isolate the effect of the optimization objective, we employed an Ordinary Least Squares (OLS) setting, excluding regularization terms and cross-validation. All other aspects, including the feature set, remained constant between approaches.

We fitted both models using 10 permutations as an observation for in-sample Mean Squared Error (MSE) comparison. A Wilcoxon signed-rank test was conducted to assess the statistical significance of the difference in MSE for Shapley value prediction between `SGUL-Shapley` and `SGUL-Accuracy`.

Table 5 presents the mean in-sample MSE on the validation set for both approaches across various datasets in the inductive setting, using the SGC model.

Table 5: In-sample validation MSE comparison between `SGUL-Shapley` and `SGUL-Accuracy` across datasets using the SGC model in the inductive setting.

| Dataset | `SGUL-Shapley` MSE | `SGUL-Accuracy` MSE | p-value |
|---------|--------------------|--------------------|---------|
| Cora | $\mathbf{1.35 \times 10^{-6}}$ | $1.40 \times 10^{-6}$ | $1.00 \times 10^{-12}$ |
| Citeseer | $\mathbf{6.08 \times 10^{-7}}$ | $6.47 \times 10^{-7}$ | $1.00 \times 10^{-12}$ |
| Pubmed | $\mathbf{2.70 \times 10^{-8}}$ | $2.73 \times 10^{-8}$ | $9.31 \times 10^{-7}$ |
| CS | $\mathbf{5.48 \times 10^{-8}}$ | $5.67 \times 10^{-8}$ | $9.77 \times 10^{-4}$ |
| Physics | $\mathbf{1.95 \times 10^{-8}}$ | $2.02 \times 10^{-8}$ | $3.13 \times 10^{-2}$ |
| Amazon-ratings | $\mathbf{3.22 \times 10^{-8}}$ | $3.26 \times 10^{-8}$ | $9.31 \times 10^{-7}$ |
| Roman-empire | $\mathbf{7.59 \times 10^{-8}}$ | $7.66 \times 10^{-8}$ | $9.31 \times 10^{-7}$ |

The results in Table 5 show that `SGUL-Shapley` consistently achieves lower MSE across all datasets, with statistically significant differences ($p < 0.05$) as determined by the Wilcoxon signed-rank test. The performance gap is particularly notable for larger datasets like CS and Physics, demonstrating `SGUL-Shapley`'s scalability and effectiveness across various graph structures.

These findings provide strong empirical support for our theoretical analysis in Section 4.2.3. They demonstrate that directly optimizing for Shapley values leads to more accurate estimation of node importance compared to the accuracy-based approach.

# J  DROPPING NODE COMPARISON OF OPTIMIZATION OBJECTIVES

To further evaluate the effectiveness of our proposed Shapley-guided optimization approach, we conducted an ablation study comparing two different objective functions: Shapley value optimization (`SGUL-Shapley`) and accuracy-based optimization (`SGUL-Accuracy`). This experiment replicates our main experimental setting and goals, as described in Section 4.2.3, but with a key difference in the optimization target for `SGUL-Accuracy`. For `SGUL-Accuracy`, we used the same form

as our proposed method but fitted it on accuracy-level data from all validation permutations. We then used this model to perform node dropping experiments, following the same procedure as in our main experiments. It's worth noting that this comparison may be influenced by factors such as cross-validation and the specific characteristics of accuracy-level data, which could introduce some variability in the results. Table 6 presents the AUC results for node dropping experiments using both methods across various datasets and models in the inductive setting.

Table 6: Comparison of AUC results for `SGUL-Shapley` and `SGUL-Accuracy` in the inductive setting. Lower AUC indicates better performance, and our proposed Shapley-guided optimization (`SGUL-Shapley`) generally outperforms the accuracy-based optimization (`SGUL-Accuracy`) in terms of AUC results across most datasets.

| | Cora | | Citeseer | | Pubmed | | CS | | Physics | | Amazon-ratings | | Roman-empire | |
|---|---|---|---|---|---|---|---|---|---|---|---|---|---|---|
| | SGC | GCN | SGC | GCN | SGC | GCN | SGC | GCN | SGC | GCN | SGC | GCN | SGC | GCN |
| SGUL-Shapley | **368.46** | **361.19** | **257.55** | 253.29 | **3068.07** | **3080.99** | **3605.92** | **3896.88** | **7797.74** | 7735.92 | **2401.77** | **2352.09** | **851.98** | 935.10 |
| SGUL-Accuracy | 369.74 | 362.66 | 257.66 | **252.87** | 3078.77 | 3090.29 | 3651.12 | 4140.17 | 7980.06 | **7707.45** | 2405.30 | 2463.65 | 852.29 | **924.64** |

The results show that `SGUL-Shapley` generally outperforms `SGUL-Accuracy` across most datasets and models. Specifically, `SGUL-Shapley` achieves better results in 10 out of 14 dataset-model combinations. The improvement is particularly noticeable for larger and more complex datasets such as CS, Physics, and Amazon-ratings. This suggests that the Shapley-guided approach is more effective at identifying critical nodes in the graph structure, especially for datasets with more intricate relationships between nodes. These findings align with our theoretical expectations and earlier in-sample error comparisons. They demonstrate that directly optimizing for Shapley values leads to more accurate and robust inference data valuation.

Table 7: Feature importance coefficients for data-specific features across datasets and models

| Feature Name | Dataset | GCN | SGC |
|---|---|---|---|
| Edge Cosine Similarity | Cora | 0 | 0 |
| | Citeseer | 0.007 | 0 |
| | Pubmed | 0 | 0.002 |
| | CS | 0.031 | 0.061 |
| | Physics | 0.003 | 0.068 |
| | Amazon-ratings | 0.025 | 0 |
| | Roman-empire | 0 | 0 |
| Representation Distance | Cora | 0.174 | 0.316 |
| | Citeseer | 0 | 0 |
| | Pubmed | 0 | 0.040 |
| | CS | 0 | 0 |
| | Physics | 0 | 0 |
| | Amazon-ratings | 0.285 | 0.032 |
| | Roman-empire | 0 | 0 |
| Classwise Rep. Distance | Cora | 0 | 0 |
| | Citeseer | 0 | 0.128 |
| | Pubmed | 0.548 | 0.383 |
| | CS | 0.195 | 0 |
| | Physics | 0 | 0 |
| | Amazon-ratings | 0 | 0 |
| | Roman-empire | 0 | 0 |

## K   ABLATION STUDY ON FEATURE CONTRIBUTIONS

To investigate the separate contributions of data-specific and model-specific features, we performed a comprehensive analysis of feature importance across different datasets and model architectures

Table 8: Feature importance coefficients for model-specific features across datasets and models

| Feature Name | Dataset | GCN | SGC |
|---|---|---|---|
| Maximum Predicted Confidence | Cora | 0 | 0 |
| | Citeseer | 0.452 | 0.291 |
| | Pubmed | 0 | 0 |
| | CS | 0.027 | 0.465 |
| | Physics | 0 | 0 |
| | Amazon-ratings | 0 | 0 |
| | Roman-empire | 0.221 | 0.273 |
| Target Class Confidence | Cora | 0.464 | 0.159 |
| | Citeseer | 0.193 | 0.230 |
| | Pubmed | 0.179 | 0.227 |
| | CS | 0.236 | 0.033 |
| | Physics | 0.738 | 0.626 |
| | Amazon-ratings | 0.298 | 0.606 |
| | Roman-empire | 0 | 0 |
| Negative Entropy | Cora | 0.194 | 0.114 |
| | Citeseer | 0.343 | 0.271 |
| | Pubmed | 0.147 | 0.176 |
| | CS | 0.147 | 0.183 |
| | Physics | 0.252 | 0.222 |
| | Amazon-ratings | 0.166 | 0.361 |
| | Roman-empire | 0.227 | 0.191 |
| Propagated Maximum Confidence | Cora | 0.168 | 0 |
| | Citeseer | 0.005 | 0.079 |
| | Pubmed | 0.110 | 0.173 |
| | CS | 0.216 | 0.258 |
| | Physics | 0 | 0.083 |
| | Amazon-ratings | 0.133 | 0 |
| | Roman-empire | 0.553 | 0.477 |
| Confidence Gap | Cora | 0 | 0.057 |
| | Citeseer | 0 | 0 |
| | Pubmed | 0 | 0 |
| | CS | 0.148 | 0 |
| | Physics | 0.007 | 0 |
| | Amazon-ratings | 0.092 | 0 |
| | Roman-empire | 0 | 0.059 |

(GCN and SGC on inductive setting). Our feature coefficients were obtained through L1-regularized optimization, where each coefficient represents the feature's contribution to the utility function. To ensure fair comparison, we normalized these coefficients within each dataset-model combination so they sum to 1, allowing us to compare relative importance across different settings.

The detailed results for data-specific features are presented in Table 7 and the detailed results for model-specific features are shown in Table 8.

To quantify the overall feature importance, we further examine the feature selection frequency. For each feature, we count its appearance (non-zero coefficient) across datasets and normalize by the total number of datasets, providing insight into how consistently each feature is selected by our L1-regularized optimization. The summary of feature selection frequencies is presented in Table 9.

The analysis reveals several key patterns in feature importance. Model-specific features exhibit higher and more consistent selection rates across datasets, with Negative Entropy being selected in all datasets and Target Class Confidence appearing in 85.7% of datasets for both architectures. This

Table 9: Feature selection frequency across datasets for GCN and SGC models

| Feature Type | Feature Name | GCN | SGC |
|---|---|---|---|
| Data-specific | Edge Cosine Similarity | 0.429 | 0.429 |
| | Representation Distance | 0.286 | 0.429 |
| | Classwise Rep. Distance | 0.286 | 0.286 |
| Model-specific | Maximum Predicted Confidence | 0.429 | 0.429 |
| | Target Class Confidence | 0.857 | 0.857 |
| | Negative Entropy | 1.000 | 1.000 |
| | Propagated Maximum Confidence | 0.714 | 0.714 |
| | Confidence Gap | 0.429 | 0.286 |

suggests that model-specific features capture fundamental aspects of model behavior independent of dataset characteristics.

On the other hand, data-specific features show more selective usage, with frequencies ranging from 0.286 to 0.429, indicating that they may be more dataset-dependent. The varying selection patterns suggest that data-specific features capture dataset-specific characteristics that complement the more universal model-specific features.

Interestingly, the selection patterns are remarkably consistent between GCN and SGC architectures, with only minor differences in selection frequencies. This consistency across architectures suggests that our feature design successfully captures fundamental aspects of graph inference quality rather than architecture-specific characteristics.

These findings support our feature design choices and demonstrate the complementary roles of data-specific and model-specific features in utility estimation. While model-specific features provide a universal basis for assessing model performance, data-specific features allow the utility learning model to adapt to the unique characteristics of each dataset. Notably, even for heterophilous graphs where GNNs typically perform worse, data-specific features like edge cosine similarity can still be selected, as shown in the ablation study tables for the Amazon-ratings. This highlights the ability of our utility learning framework to capture the nuanced relationship between graph homophily and test accuracy.

## L   LARGE-SCALE EVALUATION ON OGB-ARXIV

To demonstrate the scalability and effectiveness of our `SGUL` framework on large-scale graph datasets, we conducted additional experiments on the ogbn-arxiv dataset (Hu et al., 2020b). This dataset represents a citation network consisting of 169,343 nodes and 1,166,243 edges, where each node represents an arXiv paper and each directed edge indicates a citation.

### L.1   EXPERIMENTAL SETUP

For this experiment, we randomly sampled 10% nodes from each original train/val/test split and used 50 permutations for utility learning and 5 permutations for testing valuation. This resulted in a substantial evaluation set of over 27,000 testing neighbor nodes - to the best of our knowledge, this represents the first attempt at graph data valuation of this magnitude.

### L.2   NODE DROPPING RESULTS

Following the same evaluation protocol described in Section 6, we conducted node dropping experiments on the ogbn-arxiv dataset. The results are shown in Figure 4 and Table 10.

Table 10 presents the performance at key points during the node dropping process:

The results demonstrate `SGUL`'s superior performance on large-scale graphs. It achieves the lowest AUC score (12834.35), significantly outperforming traditional approaches like ATC-MC (12989.56)

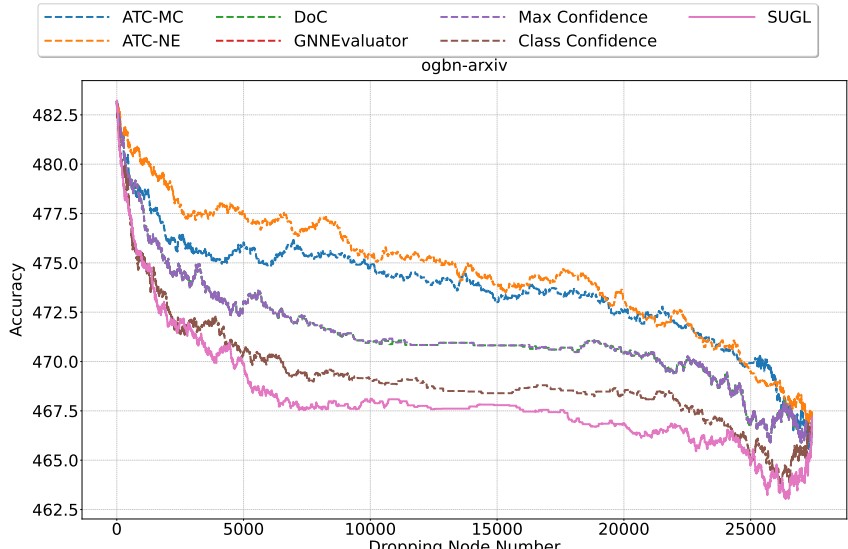

Figure 4: Accuracy curves for node dropping experiments on the ogbn-arxiv dataset using the SGC model in the inductive setting. Our proposed `SGUL` method demonstrates superior performance, achieving a steeper decline in accuracy as high-value nodes are removed.

Table 10: Performance across the node dropping process on ogbn-arxiv dataset

| Method | Start (idx 0) | 5K nodes | 10K nodes | 15K nodes | 20K nodes | End (idx 27421) | AUC |
|---|---|---|---|---|---|---|---|
| ATC-MC | 0.4832 | 0.4760 | 0.4748 | 0.4730 | 0.4726 | 0.4672 | 12989.56 |
| ATC-NE | 0.4832 | 0.4776 | 0.4755 | 0.4738 | 0.4731 | 0.4672 | 13013.93 |
| DoC | 0.4832 | 0.4730 | 0.4710 | 0.4708 | 0.4704 | 0.4672 | 12923.52 |
| Max Confidence | 0.4832 | 0.4730 | 0.4710 | 0.4708 | 0.4704 | 0.4672 | 12923.55 |
| Class Confidence | 0.4832 | 0.4705 | 0.4690 | 0.4684 | 0.4684 | 0.4672 | 12864.68 |
| SGUL | 0.4832 | **0.4698** | **0.4680** | **0.4678** | **0.4668** | 0.4672 | **12834.35** |

and ATC-NE (13013.93). Note that GNNEvaluator encounters Out-of-Memory (OOM) errors on this large dataset, further highlighting the efficiency advantage of our approach.

# M   DETAILED ALGORITHM DESCRIPTION AND EVALUATION PROTOCOL

To provide a comprehensive understanding of our framework's implementation and evaluation, we present a detailed description through three key components: (1) Shapley-Guided Utility Learning (`SGUL`) for learning utility functions from validation data, (2) Test-time Structure Value Estimation for inference without ground truth labels, and (3) Node Dropping Evaluation Protocol for empirical validation. These components collectively implement and evaluate the methodology introduced in Section 4 of our paper.

## M.1   TRAINING PHASE: SHAPLEY-GUIDED UTILITY LEARNING

Algorithm 1 implements our end-to-end optimization framework for graph inference data valuation. The algorithm takes as input a validation graph $G_{Val}$, a training graph $G_{Tr}$, a fixed trained GNN model $f(\cdot)$, the number of permutations $M$, and a regularization parameter $\lambda$. As introduced in Section 4.1, the framework begins by initializing data structures $\Psi$ and $\Phi$ to store Feature Shapley vectors and true Shapley values respectively, enabling systematic accumulation of value estimates across validation nodes.

---

**Algorithm 1** Shapley-Guided Utility Learning (SGUL)

---

**Input**: Validation graph $G_{Val} = (V_{Val}, E_{Val}, X_{Val})$, Training graph $G_{Tr}$, Fixed trained GNN model $f(\cdot)$, Number of permutations $M$, Regularization parameter $\lambda$
**Output**: Optimal parameter vector $\mathbf{w}^*$

1: **Initialize:**
2: $\quad \Psi \leftarrow \emptyset$ $\hspace{5cm}$ ▷ Feature Shapley matrix
3: $\quad \Phi \leftarrow \emptyset$ $\hspace{5.2cm}$ ▷ True Shapley values
4: **for** each node $i \in N(V_{Val})$ **do**
5: $\qquad$ Generate $M$ valid permutations $\{\pi_m\}_{m=1}^M \in \Omega(N(V_{Val}))$
6: $\qquad$ **for** each permutation $\pi_m$ **do**
7: $\qquad\qquad$ Construct subgraph sequence $\{G_{sub}(\pi_m, t)\}_{t=1}^T$
8: $\qquad\qquad$ Extract features $\mathbf{x}(S)$ for each subgraph
9: $\qquad\qquad$ Compute utility values $U(S)$ using validation accuracy
10: $\qquad$ Compute feature Shapley vector $\psi_i$:
11: $\qquad$ **for** each feature $k$ **do**
12: $\qquad\qquad \phi_i(U_k) \leftarrow \frac{1}{M} \sum_{m=1}^M \left[ U_k(N_i^{\pi_m} \cup \{i\}) - U_k(N_i^{\pi_m}) \right]$
13: $\qquad \psi_i \leftarrow [\phi_i(U_1), \phi_i(U_2), \ldots, \phi_i(U_d)]^\top$
14: $\qquad$ Compute true Shapley value $\phi_i(U)$
15: $\qquad \Psi \leftarrow \Psi \cup \{\psi_i\}$
16: $\qquad \Phi \leftarrow \Phi \cup \{\phi_i(U)\}$
17: **Optimize parameter vector:**
18: $\mathbf{w}^* \leftarrow \arg\min_{\mathbf{w}} \sum_{i \in N(V_{Val})} (\phi_i(U) - \mathbf{w}^\top \psi_i)^2 + \lambda \|\mathbf{w}\|_1$
19: **Return** $\mathbf{w}^*$

---

he core computation occurs in Step 2, implementing the feature extraction and value computation process detailed in Section 4.2.1. For each node $i$ in the validation graph neighborhood $N(V_{Val})$, the algorithm generates $M$ valid permutations through Algorithm 4 that respect graph connectivity constraints. Each permutation $\pi_m$ produces a sequence of subgraphs $\{G_{sub}(\pi_m, t)\}_{t=1}^T$ through incremental node addition. For each subgraph, we extract comprehensive features $\mathbf{x}(S)$ spanning both data-specific measures (edge cosine similarity, representation distance) and model-specific measures (confidence scores, entropy values), with validation accuracy establishing ground truth utility values $U(S)$.

Following Section 4.2.2, the algorithm constructs Feature Shapley vectors $\psi_i$ by computing individual Shapley values $\phi_i(U_k)$ for each feature type $k$. This process captures each node's contribution across multiple utility metrics, creating a rich representation incorporating both graph topology and model behavior. The framework concludes with the optimization step described in Section 4.2.3, where parameter vector $\mathbf{w}$ is optimized through our objective function to directly minimize Shapley prediction error while promoting sparsity through L1 regularization.

## M.2 INFERENCE PHASE: TEST-TIME STRUCTURE VALUE ESTIMATION

Algorithm 2 demonstrates our test-time structure valuation process. Given a test graph $G_{Te}$, target nodes $V_t$, and the learned parameter vector $\mathbf{w}^*$ from Algorithm 1, we estimate Structure-aware Shapley values for test neighbor nodes without requiring ground truth labels.

The test-time algorithm follows a similar permutation sampling and feature extraction pipeline as training, but crucially operates without access to ground truth labels. For each neighbor node $i \in N(V_t)$, we generate valid permutations using Algorithm 4 respecting graph connectivity and construct corresponding subgraph sequences. We extract the same transferable features established in the training phase, computing predicted utility $\hat{U}(S)$ through the learned linear combination $\mathbf{w}^{*\top} \mathbf{x}(S)$. These predicted utilities enable Structure-aware Shapley value estimation through permutation sampling, effectively quantifying each neighbor's contribution during inference.

Together, these algorithms form a complete framework for graph inference data valuation, enabling both efficient training of the utility function and rapid value estimation during test time. The framework's key innovation lies in its ability to learn utility functions without test labels while

maintaining computational efficiency through structured feature extraction and direct Shapley value optimization.

---

**Algorithm 2** Test-time Structure Value Estimation

---

**Input**: Test graph $G_{Te} = (V_{Te}, E_{Te}, X_{Te})$, Target nodes $V_t \subset V_{Te}$, Learned parameter vector $\mathbf{w}^*$, Number of permutations $M$, Fixed trained GNN model $f(\cdot)$

**Output**: Estimated Structure-Aware Shapley values $\{\hat{\phi}_i\}_{i \in N(V_t)}$ for test neighbor nodes

  1: **Initialize:**
  2:   $\hat{\Phi} \leftarrow \emptyset$                                           ▷ Estimated Structure-Aware Shapley values set
  3: **for** each node $i \in N(V_t)$ **do**
  4:     Generate $M$ valid permutations $\{\pi_m\}_{m=1}^M \in \Omega(N(V_t))$
  5:     **for** each permutation $\pi_m$ **do**
  6:         Construct subgraph sequence $\{G_{sub}(\pi_m, t)\}_{t=1}^T$
  7:         Extract transferable features $\mathbf{x}(S)$
  8:         Compute predicted accuracy $\hat{U}(S) = \mathbf{w}^\top \mathbf{x}(S)$
  9:     Estimate Structure-Aware Shapley value:
10:       $\hat{\phi}_i = \frac{1}{M} \sum_{m=1}^M \left[ \hat{U}(N_i^{\pi_m} \cup \{i\}) - \hat{U}(N_i^{\pi_m}) \right]$
11:       $\hat{\Phi} \leftarrow \hat{\Phi} \cup \{\hat{\phi}_i\}$
12: **Return** $\hat{\Phi}$

---

### M.3   Node Dropping Evaluation Protocol

To empirically validate the effectiveness of our structure-aware Shapley values, we conduct comprehensive node dropping experiments. This evaluation protocol measures how model performance degrades as we sequentially remove nodes ranked by their estimated importance, providing a direct assessment of our valuation framework's ability to identify critical graph structures.

Algorithm 3 outlines the node dropping evaluation process. Given a test graph $G_{Te}$, target nodes $V_t$, and the Structure-aware Shapley values $\{\hat{\phi}_i\}_{i \in N(V_t)}$ computed using Algorithm 2, we rank the nodes in descending order of their estimated importance. We then iteratively remove nodes following this ranking and measure the model's prediction accuracy after each removal. The accuracy curve and Area Under the Curve (AUC) metric are used to quantify the effectiveness of our valuation method.

A lower AUC score indicates superior performance, as it represents a more rapid and sustained decline in model accuracy when high-value nodes are removed. This aligns with our goal of accurately identifying influential graph structures - removing truly important nodes should significantly impact model performance. As demonstrated in Section 6.4, SGUL consistently achieves lower AUC scores compared to baseline methods, validating its effectiveness in identifying critical graph structures.

Furthermore, the shape of the accuracy curve provides additional insights into our method's behavior. An ideal curve should show a sharp initial drop, indicating the removal of highly influential nodes, followed by a consistent downward trend, suggesting stable and reliable importance rankings. Our experimental results in Section 6.4 demonstrate that SGUL's node dropping curves exhibit these desirable characteristics across various datasets and model architectures.

### M.4   Sampling Permissive Permutations

The structure-aware Shapley value in our framework is formally defined as:

$$\phi_i(N(\mathcal{V}_t), U) = \frac{1}{|\Omega(N(\mathcal{V}_t))|} \sum_{\pi \in \Omega(N(\mathcal{V}_t))} [U(N_i^\pi(\mathcal{V}_t) \cup \{i\}) - U(N_i^\pi(\mathcal{V}_t))]$$

This can be viewed as an expectation that we approximate through Monte Carlo sampling under precedence constraints. Specifically, we approximate the Shapley value using $M$ random permutations:

---

**Algorithm 3** Node Dropping Evaluation

---

**Input**: Test graph $G_{Te} = (V_{Te}, E_{Te}, X_{Te})$, Target nodes $V_t \subset V_{Te}$, Structure-aware Shapley values $\{\hat{\phi}_i\}_{i \in N(V_t)}$, Fixed trained GNN model $f(\cdot)$
**Output**: Accuracy curve $\{Acc_k\}_{k=1}^K$, Area Under the Curve (AUC)

1: Rank nodes in $N(V_t)$ by $\{\hat{\phi}_i\}_{i \in N(V_t)}$ in descending order
2: $K \leftarrow |N(V_t)|$
3: **for** $k = 1$ to $K$ **do**
4:    Remove top-$k$ ranked nodes from $G_{Te}$ to obtain $G_k$
5:    Compute accuracy: $Acc_k = \frac{1}{|V_t|} \sum_{v \in V_t} \mathbb{I}(f(G_k)(v) = y_v)$
6: Compute AUC: $AUC = \frac{1}{K} \sum_{k=1}^K Acc_k$
7: **Return** $\{Acc_k\}_{k=1}^K$, $AUC$

---

$$\hat{\phi}_i(N(\mathcal{V}_t), U) = \frac{1}{M} \sum_{m=1}^M [U(N_i^{\pi_m}(\mathcal{V}_t) \cup \{i\}) - U(N_i^{\pi_m}(\mathcal{V}_t))]$$

To implement this approximation while respecting graph structure constraints, we employ Algorithm 4.

---

**Algorithm 4** Precedence-Constrained Permutation Sampling

---

**Input**: Test graph $\mathcal{G} = (\mathcal{V}, \mathcal{E}, \mathbf{X})$, Target nodes $\mathcal{V}_t \subset \mathcal{V}$, Number of samples $M$, Number of hops $k$
**Output**: Set of valid permutations $\{\pi_m\}_{m=1}^M$

1: **for** $m = 1$ to $M$ **do**
2:    **Initialize:** $\mathcal{V}_{\text{visited}} \leftarrow \mathcal{V}_t$
3:    $\mathcal{V}_{\text{active}} \leftarrow \{v \in \mathcal{N}_1(\mathcal{V}_t) \mid v \notin \mathcal{V}_{\text{visited}}\}$
4:    **while** $|\mathcal{V}_{\text{active}}| > 0$ and $|\mathcal{V}_{\text{visited}}| < k$ **do**
5:       Sample $v$ from $\mathcal{V}_{\text{active}}$
6:       Update $\mathcal{V}_{\text{visited}} \leftarrow \mathcal{V}_{\text{visited}} \cup \{v\}$
7:       $\mathcal{V}_{\text{new}} \leftarrow \{u \in \mathcal{N}_1(v) \mid u \notin \mathcal{V}_{\text{visited}}\}$
8:       $\mathcal{V}_{\text{active}} \leftarrow \mathcal{V}_{\text{active}} \cup \mathcal{V}_{\text{new}} \setminus \{v\}$
9: **Return** $\{\pi_m\}_{m=1}^M$

---

This algorithm ensures each sampled permutation $\pi$ satisfies the precedence constraint by maintaining connectivity. Starting from target nodes $\mathcal{V}_t$, we iteratively sample nodes from their active neighbors (those not yet visited) while preserving graph connectivity. This sampling process guarantees that each permutation respects the structural dependencies inherent in the message-passing process, as demonstrated through our comprehensive experimental results in Section 6.

