# OpenReview forum: "Shapley-Guided Utility Learning for Effective Graph Inference Data Valuation"
_ICLR.cc/2025/Conference — ICLR 2025 Poster_

### Official Review · Reviewer_cyJv · 2024-10-28

**Soundness:** 3
**Presentation:** 3
**Contribution:** 2
**Rating:** 6
**Confidence:** 3

**Summary:**

This paper is well-written. It combines transferable data-specific and modelspecific features to approximate test accuracy without relying on G-T labels.

**Strengths:**

1. This work introduces a transferable feature extraction method that transforms player-dependent inputs into general features.

2. Authors claim that they are the first to formulate the graph inference data valuation problem.

3. Code is open-source and readable.

**Weaknesses:**

1. Lack of comparison with other works. GNNEvaluator, DoC, ATC are the compared baseline in your experiment, but readers are not yet clear about the main differences between these baselines and your method. I noticed that there is an introduction in the appendix, but there is a lack of comparison.

2. The method in this work has special optimization in structure(Sec 3.2), but lacks experimental verification of this optimization.

**Questions:**

1. Is there any experiment to show the importance of  the structure-aware Shapley value?


2. Since you are the first to formulate the graph inference data valuation problem, what's the main different between your work and GNNEvaluator[1]?It seems that both of you are methods for verifying GNN performance without  labels.

3. In your code, Which part of the paper does 'data_without_edges' correspond to? Why remove the graph structure for testing？



[1] GNNEvaluator: Evaluating GNN Performance On Unseen Graphs Without Labels, arxiv 2310.14586

---

> ### Author Response · Authors · 2024-11-24
>
> > W1. "**Lack of comparison with other works. GNNEvaluator, DoC, ATC are the compared baseline in your experiment, but readers are not yet clear about the main differences between these baselines and your method. I noticed that there is an introduction in the appendix, but there is a lack of comparison.**"
>
> > Q2. "**Since you are the first to formulate the graph inference data valuation problem, what's the main different between your work and GNNEvaluator[1]? It seems that both of you are methods for verifying GNN performance without labels.**"
>
> **A1.** Thank you for raising these important questions about the relationship between our work and existing label-free model evaluation methods. This was indeed a missing aspect in the original draft, and we have now included a detailed discussion in **Appendix D.3**. We also provide a concise summary and clarification here.
>
> Our work introduces the novel problem of graph inference data valuation, which aims to quantify the importance of individual graph structures during test time. While methods like GNNEvaluator, DoC, and ATC share our capability of operating without test labels, they address a fundamentally different objective: predicting overall model performance under distribution shifts.
>
> The key distinction lies in the problem formulation. In graph inference data valuation, we evaluate numerous subgraph configurations to measure each structure's marginal contribution to model performance, essentially decomposing the prediction process. This differs from general model evaluation where the goal is to estimate accuracy on a fixed test graph. Since our framework requires utility values for different subgraph combinations, these label-free evaluation methods can serve as utility estimators within our value assignment process, which is why we adapt them as baselines in our experiments.
>
> However, these methods are not optimized for data valuation scenarios because: (1) They focus on accuracy prediction rather than quantifying structural importance; (2) They require evaluating many subgraph permutations, leading to computational challenges (notably, GNNEvaluator encounters Out-of-Memory errors on medium-sized datasets); (3) Their architectures are designed for one-time evaluation rather than repeated utility assessment across permutations.
>
> Our SGUL framework addresses these limitations through specialized optimization techniques (Section 4.2.3) and efficient feature extraction methods (Section 4.2.1). The experimental results in Section 6.4 demonstrate that SGUL significantly outperforms the adapted baselines, validating the effectiveness of our purpose-built approach for graph inference data valuation.
>
> ---
>
> > W2: "**The method in this work has special optimization in structure(Sec 3.2), but lacks experimental verification of this optimization.**"
>
>
> **A2.** We appreciate your valuable feedback on the need for experimental validation of our proposed end-to-end optimization method. Our data valuation solution concept indeed requires special optimization, which we have validated in our experiments as detailed below.
>
> The structure-aware Shapley value formulation in Section 3.2 introduces a value assignment function incorporating graph connectivity constraints through the precedence constraint from PC-Winter [1]. This formulation quantifies each neighbor node's marginal contribution while respecting graph topology:
>
> $\phi_i(N(V_t), U) = \frac{1}{|\Omega(N(V_t))|} \sum_{\pi \in \Omega(N(V_t))} [U(N^{\pi}_i(V_t) \cup \{i\}) - U(N^{\pi}_i(V_t))]$
>
> The key challenge in our work is optimizing this formulation during test-time inference where the utility function $U(\cdot)$ cannot be directly computed due to the absence of ground truth labels for measuring model accuracy. Our novel framework (SUGL) addresses this challenge through two components: (1) A utility learning mechanism that enables test-time valuation by approximating $U
> (\cdot)$ without test labels, validated through comprehensive experiments in Section 6.4, and (2) A Shapley-guided optimization approach (which can be seen as a novel end-to-end optmization method) that directly optimizes structure-aware Shapley values, as demonstrated in Section 6.5.
>
> We believe (2) in our framework is the special optimization structure the reviewer refers to. The effectiveness of this optimization framework is empirically validated through ablation studies comparing SGUL-Shapley with accuracy-based optimization (SGUL-Accuracy). The results show SGUL-Shapley achieves superior AUC scores in 10 out of 14 dataset-model combinations, with particularly strong performance on complex datasets like CS, Physics, and Amazon-ratings.
>
> [1] Chi, Hongliang, et al. "Precedence-Constrained Winter Value for Effective Graph Data Valuation." arXiv preprint arXiv:2402.01943 (2024).

---

> ### Author Response · Authors · 2024-11-24
>
> > Q1."**Is there any experiment to show the importance of the structure-aware Shapley value?**"
> >
>
> **A3.** Thank you for asking about the experimental validation of structure-aware Shapley value. We would like to clarify that the structure-aware Shapley value formulation in our work builds upon the foundation established by PC-Winter [1] in graph data valuation. While we adopt their precedence constraint to capture connectivity dependencies, this adaptation is not the major contribution of our work. Therefore, we believe a dedicated experiment solely to demonstrate the importance of this formulation may not be necessary.
>
> As detailed in Section 3.2, we specifically focus on addressing a fundamentally different challenge: the valuation of graph structures during test-time inference, where ground truth labels are unavailable. This represents a significant departure from PC-Winter, which focuses on training data valuation where validation labels can be used to measure utility.
>
> Instead of extensively comparing different Shapley value formulations, our experimental evaluations focus on validating our key technical contributions: (1) The novel utility learning framework that enables test-time valuation without labels, as demonstrated through our comprehensive results in Section 6.4. (2) The effectiveness of our Shapley-guided optimization approach, validated through ablation studies in Section 6.5 where SGUL-Shapley shows significant improvements over SGUL-Accuracy across multiple datasets.
>
> This experimental design reflects our primary contribution of enabling graph inference data valuation in scenarios where traditional utility measurements are impossible due to the absence of test labels. The structure-aware formulation serves as a necessary foundation for this broader goal rather than being a key innovation requiring separate validation.
>
> [1] Chi, Hongliang, et al. "Precedence-Constrained Winter Value for Effective Graph Data Valuation." arXiv preprint arXiv:2402.01943 (2024).
>
> ---
>
> Q3. "**In your code, Which part of the paper does 'data_without_edges' correspond to? Why remove the graph structure for testing?**"
>
> **A4.** Thank you for asking about the implementation detail regarding 'data_without_edges'. If we understand correctly, you're referring to the code in **preprocess.py**, specifically the functions `process_inductive_planetoid_pmlp()` and `process_non_planetoid_pmlp()` where we set:
>
> ```python
> # Prepare training data (without edges for inductive setting)
> train_edge_index = torch.tensor([[],[]], dtype=torch.long).to(device)
> ```
>
> This implementation corresponds to our use of Parameterized MLPs (PMLPs) as described in Section 6.1 of our paper. As we explain in the experimental setup:"To better highlight the importance of testing structures, we employ different fixed GNN models for inductive and transductive settings. In the inductive setting, we utilize  Parameterized MLPs (PMLPs) [1], which train as standard MLPs but adopt GNN-like message passing during inference."
>
> The empty edge index during training is intentional and aligns with the PMLP design principle: during training, the model learns node representations without any graph structure (like a standard MLP), while during inference time, it leverages the graph structure through message passing operations (like a GNN). This setup helps us isolate and evaluate the importance of test-time graph structures, as the model's performance differences can be directly attributed to the graph structure used during inference.
>
> [1] Yang, Chenxiao, et al. "Graph neural networks are inherently good generalizers: Insights by bridging gnns and mlps." arXiv preprint arXiv:2212.09034 (2022).
>
> ---
> **We believe that we have responded to and addressed all your concerns and questions — in light of this, we hope you consider raising your score. Feel free to let us know in case there are outstanding concerns, and if so, we will be happy to respond.**

---

> ### Comment · Reviewer_cyJv · 2024-11-24
> **Thanks for your response**
>
> Thanks for addressing the concerns. I will consider to increase my rating. Besides, I suggest that some of the discussions in Appendix D should be placed in the main paper.

---

> > ### Author Response · Authors · 2024-11-25
> >
> > Thank you so much for your helpful suggestions! As you recommended, we’ve moved the key discussions from Appendix D to the **Related Work** section (now highlighted in blue). We’d love to hear any additional thoughts you have on the updated version. Thanks again for your support in helping us continuously improve our work!

---

### Official Review · Reviewer_KEdX · 2024-10-31

**Soundness:** 2
**Presentation:** 1
**Contribution:** 2
**Rating:** 5
**Confidence:** 4

**Summary:**

In this paper, the authors design a shapely-guided unility learning framework for graph inference data valuation, which termed SGUL. SGUL combines transferable data-specific and model-specific features to test data without relying on labels.

**Strengths:**

1. The author has provided the code to ensure the reproducibility of the paper’s results.

2. The author has supplied partial theoretical proofs to support the claims made in the study.

**Weaknesses:**

1.  In Figure 1, the author has chosen a comparison method that only includes one paper from 2024. Please incorporate more recent comparative experimental methods.


2. The results of the comparative experiments by the author are perplexing. Why does SGUL consistently perform the worst across all datasets? The author explains that SGUL can identify important nodes in the graph structure. However, could it be that the design methodology is ineffective, leading to the model’s poor performance?


3. In the field of graph neural networks, test-time training methods are widely used to address distribution shift issues in test data[1,2,3,4]. These methods also do not require labels. How does your approach compare to test-time training-based graph methods, and what advantages does it offer?


4. In Theorem 1, why is the variable U suddenly linear when it was previously described in exponential form (as in Equation 1)? The transition lacks a clear explanation, even though the author provides a proof based on the linear function.

5. There are typographical errors, such as consecutive “for instance” phrases in line 206.

6. The advantages in terms of time and space efficiency are not clearly demonstrated and need to be supported by experimental evidence.

[1] Jin W, Zhao T, Ding J, et al. Empowering graph representation learning with test-time graph transformation[J]. arXiv preprint arXiv:2210.03561, 2022

[2]Wang L, He D, Zhang H, et al. GOODAT: Towards Test-Time Graph Out-of-Distribution Detection[C]//Proceedings of the AAAI Conference on Artificial Intelligence. 2024, 38(14): 15537-15545.

[3]Zheng X, Song D, Wen Q, et al. Online GNN Evaluation Under Test-time Graph Distribution Shifts[J]. arXiv preprint arXiv:2403.09953, 2024.

[4]Pi L, Li J, Song L, et al. Test-Time Training with Invariant Graph Learning for Out-of-Distribution Generalization[J]. Available at SSRN 4886269.

**Questions:**

Please see weakness.

---

> ### Author Response · Authors · 2024-11-24
>
> > W1. "**In Figure 1, the author has chosen a comparison method that only includes one paper from 2024. Please incorporate more recent comparative experimental methods.**"
> >
>
> **A1.** Thank you for this important question regarding the selection of comparison methods. We would like to clarify that our work introduces the novel problem of graph inference data valuation, which focuses on quantifying the importance of test-time neighbors through structure-aware Shapley values. As this represents a new problem formulation, there are no direct baselines available for comparison. However, a crucial component of our solution involves predicting the testing accuracy of target nodes under different neighbor sets. Given this requirement, we innovatively bridge two research domains by adapting methods from label-free model evaluations [3,4,5] to serve as utility functions in the structure-aware Shapley value discussed in Section 3.2 as baselines.
>
>
> To establish meaningful comparisons, we carefully selected methods from the label-free evaluation domain that could be effectively adapted to our setting while meeting the computational demands of data valuation. Specifically, we have included GNNEvaluator [3] as the current state-of-the-art in label-free GNN evaluation, complemented by efficient methods such as ATC [4] and DoC [5]. These methods serve as utility functions within our framework, enabling us to predict the accuracy target nodes with different neighbor sets without requiring ground truth labels. For a detailed discussion of these baseline methods and their distinctions from our framework, we refer to the newly added section at **Appendix D.3**.
>
> The fundamental challenge in our problem setting lies in evaluating $|\Omega(N(V_t))| \times |N(V_t)|$ different subgraphs to compute Shapley values, where $\Omega(N(V_t))$ represents the set of permissible permutations and $N(V_t)$ denotes the set of neighbors. This distinguishes our task from general label-free evaluation methods. While recent works such as LEBED [1] and ProjNorm [2] offer new methodologies for testing performance prediction, these methods require model retraining during both training and testing stages. Such retraining would incur prohibitive computational costs in our scenario, where we need to evaluate numerous subgraph combinations efficiently. In response to this feedback, we have expanded our related work section to include comprehensive discussions of these methods at the additional section **Appendix D.4**. To the best of our knowledge, LEBED [1] is the only graph label-free model evaluation method published in 2024, yet it does not fit our inference data valuation scenario due to its requirement for model retraining. We welcome suggestions for additional relevant recent methods we may have overlooked.
>
> [1] Zheng, X., et al. "Online GNN Evaluation Under Test-time Graph Distribution Shifts." arXiv preprint arXiv:2403.09953 (2024).
> [2] Yu, Y., et al. "Predicting out-of-distribution error with the projection norm." International Conference on Machine Learning. PMLR, 2022.
> [3] Zheng, X., et al. "Gnnevaluator: Evaluating gnn performance on unseen graphs without labels." Advances in Neural Information Processing Systems 36 (2024).
> [4] Garg, S., et al. "Leveraging unlabeled data to predict out-of-distribution performance." arXiv preprint arXiv:2201.04234 (2022).
> [5] Guillory, D., et al. "Predicting with confidence on unseen distributions." In Proceedings of the IEEE/CVF international conference on computer vision, pp. 1134-1144, 2021.
>
> ---

---

> > ### Author Response · Authors · 2024-11-24
> >
> > > W2: "**The results of the comparative experiments by the author are perplexing. Why does SGUL consistently perform the worst across all datasets? The author explains that SGUL can identify important nodes in the graph structure. However, could it be that the design methodology is ineffective, leading to the model's poor performance?**"
> >
> >
> > **A2.** Thanks for raising this important question about our experimental results. We believe there may be a misunderstanding in the interpretation of our results, as SGUL actually demonstrates the strongest performance across datasets.
> >
> > Specifically, our experimental evaluation uses node dropping accuracy curves, where lower curves indicate better performance since they show that removing the highest-valued nodes (as identified by each method) leads to larger drops in model accuracy. Looking at Figure 1 in the main paper, we can see that SGUL (represented by the solid line) consistently achieves lower accuracy curves compared to baseline methods across all datasets. For example, in the Cora dataset, SGUL achieves both a steeper initial drop (from 0.74 to 0.68) when removing the first few hundred nodes, and maintains lower accuracy throughout the node removal process compared to baselines like ATC-MC and DoC. Similar patterns can be observed in Citeseer, where SGUL's curve remains below other methods throughout the evaluation. This superior performance is further quantified in our detailed AUC analysis presented in Table 2, where SGUL achieves the lowest AUC scores (indicating better performance) across most dataset-model combinations. This reflects that our proposed SUGL framework is able to effectively identify the vital nodes affecting the testing performance.
> >
> > To address potential confusion, we have included a more detailed explanation of the Node Dropping Evaluation Protocol in **Appendix M.3**. This section provides clarity on how the evaluation metric is designed and interpreted. We welcome any further feedback or suggestions if additional clarifications are needed.

---

> > > ### Author Response · Authors · 2024-11-24
> > >
> > > > W3. "**In the field of graph neural networks, test-time training methods are widely used to address distribution shift issues in test data[1,2,3,4]. These methods also do not require labels. How does your approach compare to test-time training-based graph methods, and what advantages does it offer**?"
> > >
> > > **A3.** We appreciate your vital question regarding the comparison between our approach and test-time training methods. We have discussed the mentioned literature in **Appendix D.5 and D.4** of the updated manuscript highlighting how our framework is different from those methods. Here, we further clarify the fundamental distinctions between our data valuation framework and these approaches:
> > >
> > > The core objective of graph inference data valuation is to quantify the contribution of individual data elements through a proper value-assignment method. The method aims to offer a flexible and general-purpose framework. Specifically, the estimated data values enable diverse downstream applications - from selecting the most valuable k nodes to maximize performance, to removing least valuable nodes for denoising, to informing data purchasing decisions. This generality and free composability of values from any data subset distinguishes our approach from test-time training methods.
> > >
> > > In contrast, test-time training methods focus solely on improving model performance through different mechanisms: GTRANS [1] performs test-time graph transformation by modifying features and structure to obtain a single optimized test graph maximizing testing performance. IGT3 [4] adapts model parameters through invariant learning to improve OOD performance. While these methods achieve performance gains, they operate on the entire graph simultaneously, assuming the presence of the whole graph, rather than evaluating the relative importance of individual elements. Our framework, however, enables flexible selection of important subgraphs under various budget constraints and objectives, from performance maximization to denoising.
> > >
> > > LEBED [3] and GOODAT [2] represent distinct approaches with different goals. While LEBED theoretically could serve as a utility function, its reliance on model retraining makes it computationally infeasible for data valuation scenarios where we must evaluate numerous subgraph combinations. GOODAT focuses on graph-level OOD detection without considering accuracy or individual node contributions, making it fundamentally different from our data valuation objective.
> > >
> > > [1] Jin, W., et al. "Empowering graph representation learning with test-time graph transformation." arXiv preprint arXiv:2210.03561 (2022).
> > > [2] Wang, L., et al. "GOODAT: Towards Test-Time Graph Out-of-Distribution Detection." AAAI Conference on Artificial Intelligence (2024).
> > > [3] Zheng, X., et al. "Online GNN Evaluation Under Test-time Graph Distribution Shifts." arXiv preprint arXiv:2403.09953 (2024).
> > > [4] Pi, L., et al. "Test-Time Training with Invariant Graph Learning for Out-of-Distribution Generalization." SSRN 4886269.
> > >
> > > ---
> > >
> > > > W4. "**In Theorem 1, why is the variable U suddenly linear when it was previously described in exponential form (as in Equation 1)? The transition lacks a clear explanation, even though the author provides a proof based on the linear function.**"
> > > >
> > >
> > >
> > > **A4.** Thank you for this question about the notation in Theorem 1. We appreciate the opportunity to clarify the relationship between our mathematical formulations.
> > >
> > > We acknowledge that the transition between Equation 1 and Theorem 1 could have been explained more clearly in the manuscript. The key point we would like to clarify is that the notation $2^{N(V_t)}$ in Equation 1 does not represent an exponential function, but rather denotes the power set (the set of all possible subsets) of $N(V_t)$, which comprises the neighborhood of target nodes.
> > >
> > > Specifically, in Section 3.2, Equation 1 establishes that $U : 2^{N(V_t)} \rightarrow \mathbb{R}$ defines a utility function mapping from the power set of $N(V_t)$ to real numbers. For instance, given $N(V_t) = \{a, b\}$, the domain $2^{N(V_t)}$ equals $\{\emptyset, \{a\}, \{b\}, \{a,b\}\}$. This notation follows standard conventions [1] in cooperative game theory for evaluating coalition values.
> > >
> > > Later in Section 4.2.3, when we introduce the linear form $U(S) = w^\top x(S)$ in Theorem 1, we are not transforming an exponential function. Rather, we are specifying a concrete implementation of the utility function while maintaining its original domain ($2^{N(V_t)}$). We selected this linear formulation for both theoretical elegance and computational efficiency, as it enables the decomposition $\phi_i(U) = w^\top\psi_i$ proved in Appendix C, establishing a direct connection between learnable parameters and predicted data values errors.
> > >
> > > We thank the reviewer for helping us identify and clarify this potential source of confusion.
> > >
> > > [1] Rozemberczki, Benedek, et al. "The shapley value in machine learning." arXiv preprint arXiv:2202.05594 (2022).

---

> > > > ### Author Response · Authors · 2024-11-24
> > > >
> > > > > W5. "**There are typographical errors, such as consecutive 'for instance' phrases in line 206.**"
> > > > >
> > > > **A5.** Thanks for catching these typographical errors. We have revised the manuscript accordingly.
> > > >
> > > > ---
> > > >
> > > > > W6. "**The advantages in terms of time and space efficiency are not clearly demonstrated and need to be supported by experimental evidence.**"
> > > > >
> > > >
> > > > **A6.** We appreciate this question about efficiency evidence of our proposed framework. We have actually provided comprehensive experimental evidence for both time and space efficiency in Section 6.5.3 of our paper, where we conducted detailed efficiency analysis comparing SGUL-Shapley with SGUL-Accuracy. As presented in Table 1 of our paper, we performed rigorous efficiency comparisons. The results demonstrate that SGUL-Shapley achieves faster training times across most datasets - for example, in Citeseer, SGUL-Shapley completes training in 0.64 seconds compared to SGUL-Accuracy's 1.54 seconds. Our memory efficiency results show that SGUL-Shapley maintains a consistently low memory usage of around 16MB across all datasets, while SGUL-Accuracy's memory usage increases substantially with dataset size. This difference is particularly pronounced for larger datasets like Amazon-ratings, where SGUL-Shapley uses only 16.59MB compared to SGUL-Accuracy's 115.62MB - an approximately 7x reduction in memory usage.
> > > >
> > > >  If additional clarification is needed, feel free to let us know!
> > > >
> > > > ---
> > > > **We believe that we have responded to and addressed all your concerns — in light of this, we hope you consider raising your score. Feel free to let us know if there are outstanding ones!**

---

> > > > ### Comment · Reviewer_KEdX · 2024-11-25
> > > > **Response to Authors**
> > > >
> > > > Thanks for the authors response to address my concerns. I’ll consider increase the score.

---

> ### Author Response · Authors · 2024-11-25
> **Thank you for your response!**
>
> Thank you so much for your response! We greatly appreciate the time you have taken to read through our replies and raise score. If there are any outstanding issues or further clarifications needed, please do not hesitate to let us know.

---

### Official Review · Reviewer_u5np · 2024-11-06

**Soundness:** 3
**Presentation:** 3
**Contribution:** 3
**Rating:** 6
**Confidence:** 4

**Summary:**

The paper proposes a novel framework called Shapley-Guided Utility Learning for the graph-structured data valuation problem. The proposed method tackle with two problems associated with graph-structured data valuation: lack of test labels and indirect optimizaiton of utility function.

**Strengths:**

1. The paper is well written and organized.
2. The method is well motivated and justified.
3. The experimental results shows promising performance.

**Weaknesses:**

1. The details of permuation sampling process is not clear. Could authors elaborate on the sampling process?
2. The proposed data-specific features, such as edge cosine similarity, appear to favor graph homophily. When applying this framework to heterophilous graphs, it raises the question of whether these features would still be effective.  How is edge cosine similarity adapted for both homophilous and heterophilous graphs? Additional discussion on this point could be valuable.
3. It seems that the ablation study on the investigation of the seperate contribution of data-specific features and model-specific feature is missisng.

**Questions:**

see weakness

---

> ### Author Response · Authors · 2024-11-24
>
> > W1. "**The details of permuation sampling process is not clear. Could authors elaborate on the sampling process?**
> "
>
> **A1.** Thank you for your question! Here, we clarify the permutation sampling process in detail:
>
> The structure-aware Shapley value is formally defined as:
>
> $$\phi_i(N(V_t), U) = \frac{1}{|\Omega(N(V_t))|} \sum_{\pi \in \Omega(N(V_t))} [U(N^{\pi}_i(V_t) \cup \{i\}) - U(N^{\pi}_i(V_t))]$$
>
> This can be viewed as an expectation:
>
> $$\phi_i(N(V_t), U) = \mathbb{E}_{\pi \sim \Omega(N(V_t))}[U(N^{\pi}_i(V_t) \cup \{i\}) - U(N^{\pi}_i(V_t))]$$
>
> We employ Monte Carlo sampling under precedence constraints to approximate this expectation, which forms the theoretical foundation of our permutation sampling process. Specifically, we can approximate the Shapley value using M random permutations:
>
> $$\phi_i(N(V_t), U) \approx \frac{1}{M} \sum_{m=1}^M [U(N^{\pi_m}_i(V_t) \cup \{i\}) - U(N^{\pi_m}_i(V_t))]$$
>
> To implement this approximation while respecting graph structure constraints, we propose:
>
>
> **Algorithm 1**: Precedence-Constrained Permutation Sampling
>
> **Input**:
> - Target nodes $V_t$
> - Graph $G=(V,E)$
> - Number of samples $M$
> - Number of hops $k$
>
> **Output**: Set of valid permutations $\Pi = \{\pi_1, ..., \pi_M\}$
>
> **For** each sample $m=1$ to $M$:
>
> 1. **Initialize**:
>    - $V_{visited} \leftarrow V_t$
>    - $\mathcal{N}_k(V_t) \leftarrow$ $k$-hop neighborhood of $V_t$ in $G$
>    - $V_{active} \leftarrow \{v \in \mathcal{N}_1(V_t) \mid v \notin V_{visited}\}$
>    - $\pi_m \leftarrow \emptyset$
>
> 2. **While** $V_{active} \neq \emptyset$ and $|V_{visited}| < |\mathcal{N}_k(V_t)|$:
>    - Sample $v \sim \text{Uniform}(V_{active})$
>    - $\pi_m \leftarrow \pi_m \cup \{v\}$
>    - $V_{visited} \leftarrow V_{visited} \cup \{v\}$
>    - **Update** $V_{active}$:
>      - $V_{new} \leftarrow \{u \in \mathcal{N}_1(v) \mid u \notin V_{visited}\}$
>      - $V_{active} \leftarrow (V_{active} \setminus \{v\}) \cup (V_{new} \cap \mathcal{N}_k(V_t))$
>
> 3. $\Pi \leftarrow \Pi \cup \{\pi_m\}$
>
> **Return** $\Pi$
>
>
> This algorithm ensures each sampled permutation $\pi$ satisfies the precedence constraint by maintaining connectivity. If additional clarification is needed, feel free to let us know!
>
> ---

---

> > ### Author Response · Authors · 2024-11-24
> >
> > > W2. "**The proposed data-specific features, such as edge cosine similarity, appear to favor graph homophily. When applying this framework to heterophilous graphs, it raises the question of whether these features would still be effective. How is edge cosine similarity adapted for both homophilous and heterophilous graphs?**"
> >
> >
> > A2. Thank you for raising this important question regarding the interplay between graph homophily and the effectiveness of our data-specific features. It’s clear that you have a deep understanding of the nuances in graph learning, and we appreciate the opportunity to clarify how our framework adapts to both homophilous and heterophilous graphs.
> >
> > Indeed, recent research has demonstrated that graph homophily impacts GNN performance [1, 2], as discussed in Appendix A; GNNs typically performing better on homophilous graphs than heterophilous ones. However, our edge cosine similarity feature serves a fundamentally different purpose than heterophilous GNN research - instead of maximizing performance on heterophilous graphs, it aims to capture how the degree of homophily correlates with test accuracy.
> >
> > Our empirical analysis supports this claim through a comprehensive feature importance study across different datasets and model architectures in the inductive setting. We performed L1-regularized optimization where each coefficient represents the feature's contribution to the utility function. To ensure fair comparison, we normalized these coefficients within each dataset-model combination to sum to 1, enabling comparison across different settings. Table 1 presents the edge cosine similarity coefficients:
> >
> > Table 1: Edge Cosine Similarity coefficients across datasets
> > | Feature Type | Feature Name | Dataset | GCN | SGC |
> > |--------------|--------------|----------|-----|-----|
> > | Data-specific | Edge Cosine Similarity | Cora | 0 | 0 |
> > | | | Citeseer | 0.007 | 0 |
> > | | | Pubmed | 0 | 0.002 |
> > | | | CS | 0.031 | 0.061 |
> > | | | Physics | 0.003 | 0.068 |
> > | | | Amazon-ratings | **0.025** | 0 |
> > | | | Roman-empire | 0 | 0 |
> >
> > The results demonstrate that edge cosine similarity remains an effective transferable feature even for heterophilous graphs. For instance, in the heterophilous Amazon-ratings dataset, edge cosine similarity receives a non-zero coefficient of 0.025 for GCN, demonstrating its utility in capturing structural information relevant to model performance. Similarly, for CS and Physics datasets which exhibit moderate homophily, the feature maintains meaningful coefficients (CS: 0.031 for GCN, 0.061 for SGC; Physics: 0.003 for GCN, 0.068 for SGC).
> >
> > The effectiveness of edge cosine similarity in heterophilous contexts can be attributed to our framework's ability to learn appropriate feature weights through L1-regularized optimization. Rather than assuming homophily as a universal indicator of performance, our model learns to appropriately weight this feature based on its predictive capabilties. The complete feature coefficient analysis, including both data-specific and model-specific features across different graph types, is presented in the **Appendix L**.
> >
> > [1] Zhu, Jiong, et al. "Beyond homophily in graph neural networks: Current limitations and effective designs." Advances in neural information processing systems 33 (2020): 7793-7804.
> >
> > [2] Li, Ting Wei, Qiaozhu Mei, and Jiaqi Ma. "A metadata-driven approach to understand graph neural networks." Advances in Neural Information Processing Systems 36 (2024).

---

> > ### Comment · Reviewer_u5np · 2024-11-26
> > **Reply**
> >
> > Thanks for your response. It clarifies my questions. I will keep my score.

---

> ### Author Response · Authors · 2024-11-24
>
> > W3. "**The ablation study on the investigation of the separate contribution of data-specific features and model-specific feature is missing.**"
> >
>
> **A3.** Thank you for your question regarding the ablation study on the separate contributions of data-specific and model-specific features. To address this, we conducted a comprehensive analysis of feature importance across various datasets and model architectures (GCN and SGC in the inductive setting). The full analysis and results are provided in **Appendix L**.
>
> To quantify the data-specific and model-specific feature importance, we examine the feature selection frequency. For each feature, we count its appearance (non-zero coefficient) across datasets and normalize by the total number of datasets, providing insight into how consistently each feature is selected by our L1-regularized optimization. Here's our summary of feature selection frequencies:
>
> | Feature Type | Feature Name | GCN | SGC |
> |--------------|--------------|-----|-----|
> | **Data-specific** | Edge Cosine Similarity | 0.429 | 0.429 |
> | | Representation Distance | 0.286 | 0.429 |
> | | Classwise Rep. Distance | 0.286 | 0.286 |
> | **Model-specific** | Maximum Predicted Confidence | 0.429 | 0.429 |
> | | Target Class Confidence | 0.857 | 0.857 |
> | | Negative Entropy | 1.000 | 1.000 |
> | | Propagated Maximum Confidence | 0.714 | 0.714 |
> | | Confidence Gap | 0.429 | 0.286 |
>
> This analysis reveals several key patterns in feature importance:
>
> First, model-specific features show higher and more consistent selection rates across datasets. Notably, Negative Entropy is selected in all datasets (frequency 1.0), and Target Class Confidence appears in 85.7% of datasets for both architectures. This suggests these features capture fundamental aspects of model behavior independent of dataset characteristics.
>
> Second, data-specific features show more selective usage (frequencies 0.286-0.429), indicating they may be more dataset-dependent. The varying selection patterns suggest these features capture dataset-specific characteristics that complement the more universal model-specific features.
>
> Third, the selection patterns are remarkably consistent between GCN and SGC architectures, with only minor differences in selection frequencies. This consistency across architectures suggests our feature design successfully captures fundamental aspects of graph inference quality rather than architecture-specific characteristics.
>
> ---
> **We believe that we have responded to and addressed all your concerns and questions — in light of this, we hope you consider raising your score. Feel free to let us know in case there are outstanding concerns, and if so, we will be happy to respond.**

---

> ### Author Response · Authors · 2024-11-27
> **Thank you for your response!**
>
> Thank you for your prior feedback and recent response. We truly appreciate your engagement with our work! We respect your decision to maintain your current score and welcome any additional thoughts you may have during the extended interaction period.

---

### Official Review · Reviewer_PSYo · 2024-11-13

**Soundness:** 3
**Presentation:** 2
**Contribution:** 3
**Rating:** 8
**Confidence:** 2

**Summary:**

The paper has studied the graph inference data valuation problem by developing a new data-driven utility function and providing theoretical insights to enable direct optimization through the Shapley value decomposition. Extensive experiments on public datasets were provided in terms of multiple evaluation protocols.

**Strengths:**

- A new definition of the structure-aware Shapley value has been introduced to facilitate graph data valuation.
- The paper has extensively discussed the limitations of applying Shapley values on graph-structured data from the utility estimation and indirect optimization perspectives, resulting in the proposed SGUL framework.
- The experiment is well designed to comprehensively access both utility estimation and data valuation, covering different graph structures, downstream tasks, and multiple evaluation protocols.

**Weaknesses:**

- While the proposed method seems novel and technically sound to me, it would be better to involve a more detailed comparison and discussion with existing works (e.g., [Chi et al., 2024]) in the main text and experimental analysis.
- It remains unclear if the proposed SGUL is scalable to large graph benchmarks, such as Open Graph Benchmark (Hu et al., 2020), due to the introduction of an $\ell_1$ penalty in eq (3).
- The methodology is somewhat hard to follow and less self-contained. An illustration framework or outline algorithm would be much helpful.

**Questions:**

While it is interesting and novel to connect utility learning and Shapley value prediction through a linear projection, the training process of the data-driven utility function and how to obtain feature Shapley values ($\psi_i$) remain unclear to me.
- How to obtain/initialize $\psi_i$?
- Can `Theorem 1` be applied to general domains other than graph-structured data?
- Is there any limitation or assumption for `Theorem 1`?

---

> ### Author Response · Authors · 2024-11-24
>
> > W1. "**While the proposed method seems novel and technically sound to me, it would be better to involve a more detailed comparison and discussion with existing works (e.g., [Chi et al., 2024]) in the main text and experimental analysis.**"
>
> **A1.**  Thank you for this valuable suggestion. Here we provide a detailed comparison with PC-Winter to highlight the key differences and innovations of our work. Additionally, a more comprehensive discussion on this comparison can be found in **Appendix D.1.1.**
>
> The proposed work builds upon and extends recent advances in graph data valuation. While PC-Winter pioneered the exploration of graph data valuation by introducing constraints to capture hierarchical dependencies, our work focuses specifically on the challenging scenario of test-time graph inference valuation, where ground truth labels are unavailable. Specifically, PC-Winter addresses training data valuation by defining hierarchical elements within computation trees as the data valuation objects (players) , applying both Level and Precedence Constraints to capture structural dependencies. In contrast, our work (Section 3.1) focuses on quantifying the importance of neighbors for test nodes during inference time. We adopt the Precedence Constraint from PC-Winter while omitting the Level Constraint, as explained in Section 3.2. This design choice reflects the distinct nature of test-time neighbor relationships, which lack the clear hierarchical groupings present in training data computation trees. The Precedence Constraint proves valuable in capturing the dependencies between nodes in the message passing process during inference.
>
> A key technical distinction lies in our approach to utility function design. While PC-Winter leverages validation accuracy as their utility measure, the absence of test labels in our setting necessitates a novel solution. As detailed in Section 4.2.1, we introduce transferable data-specific and model-specific features that can effectively approximate model performance without ground truth labels. This innovation enables the evaluation of neighbor importance during inference time.
>
> Our work complements PC-Winter by extending graph data valuation to test-time scenarios, particularly crucial for applications like real-time recommendation systems and dynamic graphs where test-time structure evaluation is essential.
>
> Here's a detailed comparison table to highlight the key differences:
> | Aspect | PC-Winter value | Structure-aware Shapley value with SUGL |
> |--------|-------------------|-----------------|
> | Valuation Target | Training graph elements | Test-time neighbors |
> | Constraints Used | Level and Precedence Constraints | Precedence only |
> | Primary Challenge | Hierarchical dependencies | No test labels |
> | Utility Function | Validation accuracy | Learned Test Accuracy |
>
> ---
>
> > W2. "**It remains unclear if the proposed SGUL is scalable to large graph benchmarks, such as Open Graph Benchmark (Hu et al., 2020), due to the introduction of an ℓ1 penalty in eq (3).**"
>
> **A2.**  Thank you for your important question about scalability! We would like to clarify that SGUL's design is specifically optimized for computational efficiency, particularly when handling large-scale graphs.
>
> Our key advantage lies in the optimization formulation. As detailed in Section 4.2.3, SGUL transforms the problem from fitting $O(|\Omega(N(V_t))| \times |N(V_t)|)$ accuracy-level data points to directly optimizing over $O(|N(V_t)|)$ Shapley values. This transformation yields three significant benefits: (1) The required training data size reduces from the product of permutations and neighbors to just the number of neighbors; (2) The memory footprint decreases proportionally as we no longer need to store accuracy values for all permutation-neighbor combinations; and (3) The optimization operates in a much lower-dimensional space, which fastern the convergence. While equation (3) includes an L1 penalty term, we can leverage well-established optimization techniques like stochastic coordinate descent [1] and proximal gradient methods [2] that are specifically designed for efficient L1-regularized optimization at scale.
>
> For context, when considering large-scale benchmarks like OGB (Hu et al., 2020), SGUL's optimization complexity depends solely on the number of nodes rather than the number of permutations or edge combinations. As demonstrated in our efficiency analysis (Section 6.5.3), SGUL maintains consistent memory usage regardless of dataset size, suggesting its potential applicability to larger graph benchmarks.
>
>
> [1] Liu, Ji, et al. "An asynchronous parallel stochastic coordinate descent algorithm." International Conference on Machine Learning. PMLR, 2014.
>
> [2] Polson, Nicholas G., James G. Scott, and Brandon T. Willard. "Proximal algorithms in statistics and machine learning." (2015): 559-581.
>
> ---

---

> ### Author Response · Authors · 2024-11-24
>
> > W3. "**The methodology is somewhat hard to follow and less self-contained. An illustration framework or outline algorithm would be much helpful.**"
>
> **A3.** Great suggestion about methodology clarity! Per your advice, we have added detailed algorithm descriptions about (1) Shapley-Guided Utility Learning Algorithm, (2) Test-Time Structure-Aware Shapley Value Estimation Algorithm and (3) dropping node evaluation protocols in **Appendix M** to provide a comprehensive understanding of our framework.
>
> Here, we present the core algorithms about Shapley-Guided Utility Learning and Test-Time Structure-Aware Shapley Value Estimation:
>
> **Algorithm 1**: Shapley-Guided Utility Learning (SGUL)
>
> **Input**:
> - Validation graph $G_{Val} = (V_{Val}, E_{Val}, X_{Val})$
> - Training graph $G_{Tr}$
> - Fixed trained GNN model $f(\cdot)$
> - Number of permutations $M$
> - Regularization parameter $\lambda$
>
> **Output**: Optimal parameter vector $\mathbf{w}^*$
>
> 1. **Initialize**:
>    - $\Psi \leftarrow \{\}$ # Feature Shapley matrix
>    - $\Phi \leftarrow \{\}$ # True Shapley values
>
> 2. **For** each node $i \in N(V_{Val})$:
>    - Generate $M$ valid permutations $\\{\pi\_m\\}\_{m=1}^M \in \Omega(N(V_{Val}))$
>    - **For** each permutation $\pi_m$:
>      - Construct subgraph sequence $\{G_{sub}(\pi_m,t)\}_{t=1}^T$
>      - Extract features $\mathbf{x}(S)$ for each subgraph
>      - Compute utility values $U(S)$ using validation accuracy
>    - Compute feature Shapley vector $\psi_i$:
>      - **For** each feature $k$:
>        - $\phi_i(U_k) \leftarrow \frac{1}{M}\sum_{m=1}^M[U_k(N^{\pi_m}_i \cup \{i\}) - U_k(N^{\pi_m}_i)]$
>      - $\psi_i \leftarrow [\phi_i(U_1), \phi_i(U_2), ..., \phi_i(U_d)]^\top$
>    - Compute true Shapley value $\phi_i(U)$
>    - $\Psi \leftarrow \Psi \cup \{\psi_i\}$
>    - $\Phi \leftarrow \Phi \cup \{\phi_i(U)\}$
>
> 3. **Optimize parameter vector**:
>    - $\mathbf{w}^* \leftarrow \arg\min_{\mathbf{w}} \sum_{i \in N(V_{Val})} (\phi_i(U) - \mathbf{w}^\top\psi_i)^2 + \lambda\|\mathbf{w}\|_1$
>
> **Return**: $\mathbf{w}^*$
>
> The algorithm implements our end-to-end optimization framework for graph inference data valuation. It processes a validation graph and trained GNN model to accumulate Feature Shapley vectors and true Shapley values systematically. For each validation node, the algorithm generates permutations respecting graph connectivity, extracts comprehensive features from resulting subgraphs, and computes Shapley values capturing structural importance. The framework concludes by optimizing parameters through L1-regularized objective function to enable efficient test-time value estimation.
>
> **Algorithm 2**: Test-time Structure Value Estimation
>
> **Input**:
> - Test graph $G_{Te} = (V_{Te}, E_{Te}, X_{Te})$
> - Target nodes $V_t \subset V_{Te}$
> - Learned parameter vector $\mathbf{w}^*$
> - Number of permutations $M$
> - Fixed trained GNN model $f(\cdot)$
>
> **Output**: Estimated Structure-Aware Shapley values $\\{\hat{\phi}\_i\\}\_{i \in N(V\_t)}$ for test neighbor nodes
>
> 1. **Initialize**:
>    - $\hat{\Phi} \leftarrow \{\}$ # Estimated Shapley values
>
> 2. **For** each node $i \in N(V_t)$:
>    - Generate $M$ valid permutations $\\{\pi\_m\\}\_{m=1}^M \in \Omega(N(V_t))$
>    - **For** each permutation $\pi_m$:
>      - Construct subgraph sequence $\{G_{sub}(\pi_m,t)\}_{t=1}^T$
>      - Extract transferable features $\mathbf{x}(S)$
>      - Compute predicted accuracy $\hat{U}(S) = \mathbf{w}^{*\top}\mathbf{x}(S)$
>    - Estimate Shapley value:
>      $\hat{\phi}\_i = \frac{1}{M}\sum\_{m=1}^M[\hat{U}(N^{\pi_m}\_i \cup \{i\}) - \hat{U}(N^{\pi\_m}\_i)]$
>    - $\hat{\Phi} \leftarrow \hat{\Phi} \cup \{\hat{\phi}_i\}$
>
> **Return**: $\hat{\Phi}$
>
> This algorithm demonstrates test-time structure valuation using the learned utility function. For each test neighbor node, it generates valid permutations, constructs subgraph sequences, and extracts transferable features to estimate Structure-aware Shapley values without requiring ground truth labels.

---

> ### Author Response · Authors · 2024-11-24
>
> >Q1. "**How to obtain/initialize $\psi_i$ ?**"
> >
> **A4.**  Thank you for this important question! We here clarify how we obtain feature/ground-truth accuracy Shapley values and how to learn the utility function with our method:
>
> The training process builds upon the structure-aware Shapley formulation defined in Section 3.2. On the validation graph $G_{Val}$, we first generate a set of permissible permutations $\Omega(N(V_{Val}))$ that satisfy the precedence constraints. For each permutation $\pi \in \Omega(N(V_{Val}))$, we construct a sequence of subgraphs $\{G_{sub}(\pi,t)\}^T_{t=1}$, where $G_{sub}(\pi,t)$ contains the first $t$ nodes according to permutation $\pi$.
>
> For each subgraph $G_{sub}$, we compute two essential components: (1) First, we extract the transferable features $\mathbf{x}(S) \in \mathbb{R}^d$ as described in Section 4.2.1, where $S$ represents the set of nodes in $G_{sub}$. These features include both data-specific measures (such as edge cosine similarity and representation distance) and model-specific measures (such as prediction confidence and entropy). (2) Second, we calculate the ground truth utility $U(S)$ by measuring the model's accuracy on the validation nodes $V_{Val}$ using the subgraph structure. This provides our training pairs $\{\mathbf{x}(S), U(S)\}$.
>
> To compute Feature Shapley values $\psi_i$, we decompose the utility function into individual feature components. For each feature $k$, we define $U_k(S) = x_k(S)$ and estimate $\phi_i(U_k)$ with $M$ permutation samples as mentioned in **Algorithm 1** in the prior answer. The Feature Shapley vector $\psi_i$ is then constructed as:
>
> $$\psi_i = [\phi_i(U_1), \phi_i(U_2), ..., \phi_i(U_d)]^\top$$
>
> ---
> > Q2. "**Can Theorem 1 be applied to general domains other than graph-structured data? Is there any limitation or assumption for Theorem 1?**"
>
> **A5.**  Thank you for this great question! Theorem 1 extends beyond graph-structured data to any cooperative game setting where value functions can be defined through permutations. The theorem's generality stems from the fundamental linearity axiom for a solution concept $\phi$, which states that for any characteristic functions $v$, $w$ and scalars $\alpha$, $\beta$:
>
> $\phi_i(\alpha v + \beta w) = \alpha \phi_i(v) + \beta \phi_i(w)$
>
> For any permutation-based solution concept where the value is computed as:
>
> $\phi_i(v) = \frac{1}{|\Pi'|} \sum_{\pi \in \Pi'} [v(S_\pi(i) \cup \{i\}) - v(S_\pi(i))]$
>
> where $\Pi'$ is the set of permissible permutations, given a linear utility function $U(S) = \mathbf{w}^\top\mathbf{x}(S)$, we have:
>
> $\phi_i(U) = \frac{1}{|\Pi'|} \sum_{\pi \in \Pi'} [\mathbf{w}^\top\mathbf{x}(S_\pi(i) \cup \{i\}) - \mathbf{w}^\top\mathbf{x}(S_\pi(i))]$
> $= \mathbf{w}^\top[\frac{1}{|\Pi'|} \sum_{\pi \in \Pi'} (\mathbf{x}(S_\pi(i) \cup \{i\}) - \mathbf{x}(S_\pi(i)))]$
> $= \mathbf{w}^\top\psi_i$
>
> This decomposition applies to many important frameworks in cooperative game theory and machine learning. For instance, classical Shapley values in cooperative games, SHAP values [1] for model interpretation, Data Shapley [2] for dataset valuation, PC-Winter value [3] for graph structures and and semi-values (e.g., the Banzhaf value) [4] all share this fundamental structure. Each can be viewed as special cases where the feature vector $\mathbf{x}(S)$ captures the relevant characteristics of subset $S$ in their respective domains.
>
> [1] Lundberg, Scott. "A unified approach to interpreting model predictions." arXiv preprint arXiv:1705.07874 (2017).
>
> [2] Ghorbani, Amirata, and James Zou. "Data shapley: Equitable valuation of data for machine learning." International conference on machine learning. PMLR, 2019.
>
> [3] Chi, Hongliang, et al. "Precedence-Constrained Winter Value for Effective Graph Data Valuation." arXiv preprint arXiv:2402.01943 (2024).
>
> [4] Wang, Jiachen T., and Ruoxi Jia. "Data banzhaf: A robust data valuation framework for machine learning." International Conference on Artificial Intelligence and Statistics. PMLR, 2023.

---

> ### Author Response · Authors · 2024-11-24
>
> ---
> > Q3. **Is there any limitation or assumption for Theorem 1?**
> >
>
> **A6.** Despite Theorem 1 being applicable to a broad class of permutation-defined solution concepts—such as the classical Shapley value (with applications including SHAP and Data Shapley), the PC-Winter value, and semi-values (e.g., the Banzhaf value) in cooperative game theory, all of which adhere to the property of linearity—it does have specific limitations.
>
> Particularly, it is important to note that the direct optimization enabled by Theorem 1 specifically applies to learning parameters $\mathbf{w}$ in **linear utility learning models** of the form $U(S) = \mathbf{w}^\top\mathbf{x}(S)$. This linear structure is crucial for the decomposition. For non-linear utility functions—such as those involving neural networks, kernel methods, or other complex transformations—this direct optimization approach may not be applicable. In such cases, alternative optimization strategies would be required.
>
> ---
> **We believe that we have responded to and addressed all your concerns — in light of this, we hope you consider raising your score. Feel free to let us know if there are outstanding ones!**

---

> ### Author Response · Authors · 2024-11-29
> **New exciting results on OGB-arxiv**
>
> Hi Reviewer PSYo, thank you for your helpful feedbacks! Thanks to the new extended discussion period, we're able to finish the experiment and share our most recent experimental results addressing your concerns about SGUL's scalability on large graph benchmarks like OGB (**weakness 2**).
>
> Following your suggestion, we conducted experiments on the **ogbn-arxiv** dataset, randomly sampling 10% nodes from each original train/val/test split (with 50 permutations for utility learning and 5 permutations for testing valuation). This results in a substantial evaluation set of over 27,000 testing neighbors - **the first attempt at graph data valuation of this large magnitude** as the best knowledge of us.
>
> Following the same evaluation protocol in Section 6.4, we conducted node dropping experiments. Since we cannot update the PDF at this stage, we provide the detailed results here, which will be included in the revised version as an additional figure:
>
> Performance across dropping process (Section 6.4.1):
> | Method | Start (idx 0) | 5K nodes | 10K nodes | 15K nodes | 20K nodes | End (idx 27421) |
> |--------|--------------|-----------|------------|------------|------------|-----------------|
> | ATC-MC | 0.4832 | 0.4760 | 0.4748 | 0.4730 | 0.4726 | 0.4672 |
> | ATC-NE | 0.4832 | 0.4776 | 0.4755 | 0.4738 | 0.4731 | 0.4672 |
> | DoC | 0.4832 | 0.4730 | 0.4710 | 0.4708 | 0.4704 | 0.4672 |
> | Max Confidence | 0.4832 | 0.4730 | 0.4710 | 0.4708 | 0.4704 | 0.4672 |
> | Class Confidence | 0.4832 | 0.4705 | 0.4690 | 0.4684 | 0.4684 | 0.4672 |
> | **SGUL** | **0.4832** | **0.4698** | **0.4680** | **0.4678** | **0.4668** | **0.4672** |
>
> We also report overall Area Under Curve (AUC) Scores (as we have in Section 6.4.2):
> | Method | AUC |
> |--------|-----|
> | ATC-MC | 12989.56 |
> | ATC-NE | 13013.93 |
> | DoC | 12923.52 |
> | Max Confidence | 12923.55 |
> | Class Confidence | 12864.68 |
> | **SGUL** | **12834.35** |
>
> This encouraging results demonstrate SGUL's strong capabilities on performing data valuation on large-scale graphs. **Not only does SGUL achieve the lowest AUC score (12834.35), significantly outperforming traditional approaches like ATC-MC (12989.56) and ATC-NE (13013.93), but it also maintains consistent performance while processing this extensive evaluation set**. This also shows that **SGUL also is efficiently handling graph data valuation tasks at scale, validating that the L1 penalty in equation (3) effectively supports rather than hinders scalability.** The empirical findings complement our theoretical analysis in Section 4.2.3, showing SGUL's strong performance on large graph benchmarks.  **We greatly appreciate your question that helped enhance our evaluation.**
>
> As the discussion period between authors and reviewers nears its end, we also wanted to check in to ensure our responses have addressed your questions with this opportunity. If anything remains unclear or if you have any concerns, please don’t hesitate to reach out to us!

---

> > ### Comment · Reviewer_PSYo · 2024-12-01
> > **Post-Rebuttal Feedback**
> >
> > Thanks for providing the new results. After reading the author's rebuttal and all the other review comments, most of my previous concerns have been well addressed, especially regarding the large-scale evaluation. Thus, the reviewer will increase the rating accordingly.

---

> > > ### Author Response · Authors · 2024-12-01
> > > **Thank you for your response!**
> > >
> > > Thanks so much for your feedback and insightful questions, which have played a crucial role in enhancing our work. We deeply appreciate your valuable reviews.

---

### Author Response · Authors · 2024-11-25
**Thank You All and Paper Updates**

We appreciate all reviewers for their valuable feedback. In response to your comments:

1. We've expanded the discussion on  **label-free model evaluation methods, retraining-based methods, and test-time training & augmentation methods** in **Appendices D.3, D.4, and D.5**. A summary of these discussions has been added to the **Related Work** section in the main text.

2. To better understand feature contributions, we've included coefficient tables (**Table 1 in Appendix L**) and also conducted an ablation study (**Appendix L**), showing how data-specific and model-specific features contributes to our utility learning framework.

3. We've provided a description of the training process for our accuracy-based variant and baselines in **Appendix B**.

4. To clarify our methodology, we've introduced four algorithms in **Appendix M**:

   - Algorithm 1: Shapley-Guided Utility Learning (our proposed SUGL method)

   - Algorithm 2: Test-time Structure Value Estimation (estimate structure-aware Shapley values/testing neighbor data values)

   - Algorithm 3: Node Dropping Evaluation Protocol (experimental validation protocol)

   - Algorithm 4: Precedence-Constrained Permutation Sampling (generate valid permutations)

5. We have carefully revised the manuscript to enhance readability and correct typos.

These updates aim to make our work more accessible. We sincerely thank you for pointing out these issues.

 **We welcome your further feedback on our rebuttal. Thank you for your time and insights in helping us improve this work！**

---

### Meta-Review · Area_Chair_tChs · 2024-12-18

**Metareview:**

This paper presents a new Shapley-Guided Utility Learning for the graph-structured data valuation problem. Reviewers agreed the paper is well written and clearly organized. The paper introduces a new definition of the structure-aware Shapley value, which will be helpful for graph data valuation. In addition, the experiments are well designed, and the results are extensive and convincing. Meanwhile, reviewers raised some concerns about scalability, details of sampling process, missing ablation studies, and complexity analysis. The authors have provided detailed responses to address these concerns during the rebuttal and discussion period.

**Additional Comments On Reviewer Discussion:**

Reviewers raised some concerns about technical details and experiments, which have been sufficiently addressed by authors during the rebuttal.

---

### Decision · Program_Chairs · 2025-01-22

Accept (Poster)